



# Coupled Wind Turbine Design and Layout Optimization with Non-Homogeneous Wind Turbines

Andrew P.J. Stanley[1] and Andrew Ning[1]

[1]Department of Mechanical Engineering, Brigham Young University, Provo, UT, 84602, USA

**Correspondence:** Andrew PJ Stanley (stanley_andrewpj@byu.net)

**Abstract.** In this study, wind farms were optimized to show the benefit of coupling turbine design and layout optimization, as well as including different turbine designs in a single wind farm. For our purposes, turbine design included hub height, rotor diameter, rated power, tower diameter, tower shell thickness, and implicit blade chord and twist distributions. A 32-turbine wind farm and a 60-turbine wind farm were both considered, as well as a variety of turbine spacings and wind shear exponents. Structural constraints as well as turbine costs were considered in the optimization. Results indicate that coupled turbine design and layout optimization is superior to sequentially optimizing turbine design, then turbine layout. Coupled optimization results in an additional 2–5% reduction in cost of energy compared to optimizing sequentially. Furthermore, wind farms with closely spaced wind turbines can greatly benefit from non-uniform turbine design throughout the farm. Some of these wind farms with heterogeneous turbine design have an additional 10% cost-of-energy reduction compared to wind farms with identical turbines throughout the farm.

## 1 Introduction

Mitigating wake interactions among wind turbines is one of the most difficult challenges in wind farm design. Upstream turbines remove energy from the wind, decreasing the energy available to the rest of the farm. These wake losses often reduce the power production by 10–20% when compared to unwaked conditions (Barthelmie et al., 2007, 2009; Briggs, 2013). Thus, a major part of wind farm design is predicting and reducing wake interactions among turbines. In this paper, we minimized the cost of energy (COE) of wind farms through layout and turbine design optimization. We gave special attention to coupled design and layout optimization, and to wind farms with non-homogeneous turbine designs. To successfully optimize the many variables that come from coupling layout and turbine design, we used exact analytic gradients as opposed to one of the gradient-free optimization methods commonly used in wind farm design. Although multi-modal design spaces, like wind farm design spaces, are often well suited for gradient-free algorithms, gradient-based optimization methods can be useful in some cases, such as when using many turbines or when considering more design variables than just turbine layout. Even though gradient-free algorithms may be superior in finding global optima compared to gradient-based methods, as the number of design variables in a problem increases, the computational expense for gradient-free optimization methods rises dramatically. For large wind farms gradient-free methods become infeasible, and while gradient-based optimization methods converge to local minima, they scale much better with the number of design variables. When considering several design variables or wind



farms with many turbines, gradient-based optimization with multiple starting points becomes the best, and often only feasible solution method. Rather than limit ourselves to the 9–25 turbines typically used in gradient-free optimization studies, we used gradient-based methods to optimize wind farms of 32–60 wind turbines (with the ability to do more), coupled with as many as 18 additional variables for turbine design.

Three main methods exist to decrease wake interactions among wind turbines in a wind farm: layout optimization, yaw control, and turbine design. The wind farm layout optimization problem has been widely studied in recent years. There is abundant literature from the research community discussing various methods to approach the wind farm layout optimization problem including gradient-free methods (Marmidis et al., 2008; Emami and Noghreh, 2010; Kusiak and Song, 2010; Ituarte-Villarreal and Espiritu, 2011; Feng and Shen, 2015; Gao et al., 2015) and gradient-based methods (Pérez et al., 2013; Park

and Law, 2015; Fleming et al., 2016b; Guirguis et al., 2016; Gebraad et al., 2017). The premise of layout optimization is simple: design the wind farm layout such that wake interactions among turbines are minimal. However, the problem is more challenging than it may initially seem. The space of a wind farm is constrained, so for all realistic wind roses, any turbine layout will have some wind turbines that are waked or partially waked some or all of the time. Therefore, to find the best layout often non-obvious tradeoffs must be made to minimize wake interactions throughout the entire farm. Also, the number of wake

simulations to model a wind farm scales with the square of the number of turbines, becoming computationally expensive for farms with many turbines. Another challenge comes from the extreme multi-modality of the design space. For farms with many wind turbines, it becomes impossible to know if a solution is the global optimal solution or just a local optimum. Additional complexity arises from the stochastic nature of wind. Although often treated as deterministic, annual wind direction and speed distributions are uncertain and variable, meaning that the optimal wind farm layout for one year may not be optimal the next.

Wake steering through turbine yaw control is another method to decrease wake interactions between wind turbines (Fleming et al., 2016b; Gebraad et al., 2017). Although not considered in this paper, yaw control can be applied to the wind farms in this study for additional improvements.

     The third method to decrease wake interactions in a wind farm is turbine design. Turbine design is admittedly a broad category, involving a variety of elements. In this paper we specifically explored heterogeneous hub heights, rotor diameters,

turbine ratings, and tower diameters, tower shell thicknesses, and blade chord and twist distributions in the same wind farm. In all, these variables represent a significant portion of wind turbine design and approach complete turbine design. In recent years heterogeneous turbine design has begun to receive attention from the research community, and several studies have begun to look into wind farms with mixed turbine designs. Chen et al. optimized a wind farm layout and allowed turbines of different hub heights, finding a power output increase of 13.5% and a COE decrease of 0.4% (Chen et al., 2013). Chowdhury et al. found

a 13.1% increase in power generation in a wind farm with rotor diameter and layout treated as design variables, compared to a wind farm with just optimized layout (Chowdhury et al., 2010). In another study, Chowdhury et al. found that the capacity factor of a wind farm increases by 6.4% when the farm is simultaneously optimized for layout and turbine type, with different turbine types in the wind farm, compared to a farm where every turbine is identical (Chowdhury et al., 2013). Chen et al. also performed a study in which the layout and turbine types are optimized in a wind farm. They found that the optimal wind farms

had several different turbine types rather than one type throughout the entire farm (Chen et al., 2015). In our previous work,





we found that wind farms with low wind shear, closely spaced wind turbines, and small rotor diameters can greatly benefit from having turbines with different hub heights. For many of the wind farms that we optimized, wind farms with two different heights had an optimal COE that was 5–10% lower than the wind farms with all identical turbine heights (Stanley et al.).

The results of these studies indicate that in many situations, mixing different hub heights, rotor diameters, and turbine types increases the power production in a wind farm and decreases the COE. This paper builds on these studies mentioned and others like them. We made the following contributions, which are either novel in the field or significant improvements on previous studies. First, we included many aspects of turbine design as design variables coupled with turbine layout, rather than select one or two aspects of design or choose from a set of existing turbine models. This allowed us to fully explore the design space and discover additional benefits associated with coupled design optimization. Second, in this paper we included the cost and structural impacts of changing the turbine design in our optimization objective and constraints. Third, we used gradient-based optimization with exact analytic gradients for every aspect of our wind farm model. This allowed us to optimize large wind farms and include many design variables, which would be impossible with a gradient-free optimization approach. Fourth, we analyzed many different wind farm sizes and wind conditions. Fifth, we specifically addressed how sequentially optimizing turbine design then layout compares to fully coupling the design variables.

## 2 Methodology

### 2.1 Wake Model

We used the FLORIS wake model to predict the wind speeds throughout the wind farms in our study (Gebraad et al., 2016). The FLORIS model had some discontinuities in the original formulation, so in this study we used a version that has been modified to be smooth and continuously differentiable, enabling gradient-based optimization (Thomas et al., 2017).

The total velocity deficit at any given point was defined as the square root of the sum of the squares of the loss contribution from each turbine wake:

$$L = \sqrt{\sum_{i=1}^{\text{nTurbs}} L_i^2} \tag{1}$$

Variations of the free stream wind speed with height were calculated with the wind profile power law:

$$V = V_{\text{ref}} \left( \frac{z}{z_{\text{ref}}} \right)^{\alpha} \tag{2}$$

where $V$ is the wind speed at height $z$; $V_{\text{ref}}$ is the reference wind speed given by the wind data; $z_{\text{ref}}$ is height at which the reference wind speed was measured, which we assumed to be 50 meters; and $\alpha$ is the wind shear exponent, which defines how the wind speed varies with height.



## 2.2 Annual Energy Production Calculation

### 2.2.1 Power Calculation

We assumed that up to rated power, the rotation of each wind turbine could be controlled such that a constant power coefficient of 0.42 was achieved. The wind turbine power generation was defined as:

$$P = \begin{cases} C_P \frac{1}{2} \rho V_{\text{eff}}^3 A & V_{\text{eff}} \leq V_{\text{rated}} \\ P_{\text{rated}} & V_{\text{eff}} > V_{\text{rated}} \end{cases} \tag{3}$$

Where $C_P$ is the power coefficient; $\rho$ is the air density which we assumed was 1.1716 kg m$^{-3}$; $A$ is the swept area of the turbine rotor; and $V_{\text{eff}}$ is an effective wind speed across rotor, which was defined as:

$$V_{\text{eff}} = V(1 - L) \tag{4}$$

Where $V$ is the free stream wind speed at the turbine hub height, and L is the total velocity deficit.

### 2.2.2 Wind Speed Distributions

We represented the speeds at any wind direction as a Weibull distribution, which is commonly used to represent wind speed distributions (Justus et al., 1978; Rehman et al., 1994; Dorvlo, 2002):

$$W(x) = \left(\frac{k}{V_{\text{mean}}}\right)\left(\frac{x}{V_{\text{mean}}}\right)^{k-1} \exp\left[\left(-\frac{x}{V_{\text{mean}}}\right)^k\right] \tag{5}$$

The shape factor, $k$, was set as 1.76. The mean speed for a given distribution, $V_{\text{mean}}$, could be different depending on the wind direction, meaning that each wind direction had an associated Weibull curve defining the wind speed distribution from that direction. Figure 1 shows the wind speed Weibull distributions for two different $V_{\text{mean}}$ values.

### 2.2.3 Sampling

The direction data we had was binned into 36 directions for one wind rose, and 72 directions for the other. This is very fine sampling; from a convergence study, we found that it is more refined than necessary to accurately compute the annual energy production (AEP) of a wind farm. For every wind direction at which the power was computed, the wake model needed to be called; therefore, reducing the number of directions at which the wind farm power was computed reduced the time required to optimize. However, too few directions would make the AEP calculation inaccurate. We fit a spline to the direction data and were thus able to sample at any direction. We then performed a two-dimensional convergence study to find how many directions and speeds were required to approach the "true" AEP, which we defined to be the AEP calculated when using 50 wind directions and 30 wind speed samples. We found that at 23 wind direction samples and 5 wind speed samples from the Weibull distributions, the AEP converged within 2% of the true AEP. This was within the error of our wake model; therefore, this was the number of samples used in our study.





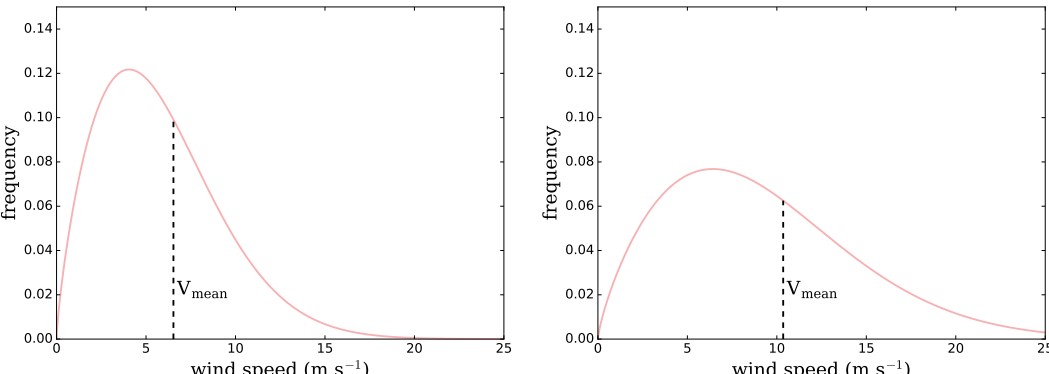

**Figure 1.** The Weibull wind speed distributions for two different average wind speeds. In (a) there is an average wind speed of 6.53 meters per second, and (b) shows an average wind speed of 10.35 meters per second. The shape factor k in each Weibull distribution was chosen as 1.76.

## 2.3 Tower Model

Because the tower height varied in this study, it was necessary to calculate the tower mass and perform structural analyses. The structural analysis was used to constrain the tower from stress or buckling issues. It was necessary to provide a model with gradients for all of our constraints, which included the von Mises stress, shell buckling, and global buckling at any point along the tower; the tower taper ratio; and the first natural frequency of the structure. The method by which these calculations were made is discussed in more detail in our previous study (Stanley et al.).

## 2.4 Rotor/Nacelle Models

The variable rotor diameter, turbine power rating, and blade and chord distributions in this study also needed be accounted for in structural analysis. To do so, we used a model developed at NREL called RotorSE to calculate the rotor mass, rated and extreme thrust, rated torque, rated wind speed, and moments of inertia (Ning, 2013a). The complex nature of RotorSE allows the user to fully define a rotor and perform analysis, however this comes at a cost in computation time. Because we coupled turbine design and turbine layout optimization, the rotor analysis needed to be called many more times than in an isolated turbine design optimization. Thus, to speed up the rotor calculations in our optimization, we created a surrogate model on the results provided by RotorSE. We sampled rotor diameters evenly spaced from 46 meters to 160 meters, every six meters, and rated powers from 0.5 megawatts to ten megawatts, every 0.5 megawatts. The lower limits, 46 meter rotor diameter and 500 kilowatt rated power, are both lower than we expected any of the optimal values to be. The upper limits, 160 meter rotor diameter and 10 megawatt rated power are both near the upper limit of current wind turbine technology. For each combination of rotor diameter and rated power, we used RotorSE to minimize the blade mass using the blade chord and twist distributions as design variables. The optimization was constrained such that the turbine blades would not fail from stress or buckling and the power coefficient was greater than 0.42. Note that we did not vary the airfoils in the optimizations. We then used the converged



optimizations, and used k-fold cross-validation with 10 groups to choose a fifth order bivariate spline which was then applied to each of the outputs of interest. This spline function was then used in place of RotorSE in our wind farm optimizations. By creating the surrogate, we achieved the accuracy of RotorSE without the large associated time requirement, as well as fast and simple analytic gradients. The k-fold cross-validation with 10 groups showed that the mean error is below 4% for the moments

5    of inertia, approximately 4.5% for the extreme thrust, and below 3% for the rest of the fits. Figure 2 shows the normalized surface fits for each of the variables of interest.

**Figure 2.** The spline fits to optimized RotorSE data. These fits were used to obtain the desired outputs of rotor mass, rated and extreme thrust, rated torque, rated wind speed, and moments of inertia as functions of the rotor diameter and rated power.





## 2.5 Cost Model

AEP is a standard objective in wind farm optimization problems because it is easy to calculate and is a valid measure when only power production is affected by the optimization. When aspects of turbine design are included as design variables, this measure is no longer appropriate because of costs of the wind farm are effected as well. To accurately represent the trade-offs
between power production and cost, we evaluated our wind farm by its COE as was done in our previous paper on wind farms with different turbine heights (Stanley et al.).

## 2.6 Optimization

We set up our optimization with two different turbine groups. We assigned each turbine to one of two groups, where all turbines in a group had the same tower hub height, rotor diameter, turbine rating, tower diameter, tower shell thickness, and
blade chord and twist distributions. Rather than optimize each turbine, we chose two groups because our previous study in which we optimized wind farms with different turbine heights indicated that the most benefit comes from increasing from one height group to two. Any benefit from introducing more groups was marginal (Stanley et al.). We parameterized the tower by specifying the diameter and shell thickness at the bottom, midpoint, and top of the tower and then linearly interpolating diameter and shell thickness at points in between, as shown in Fig. 3.

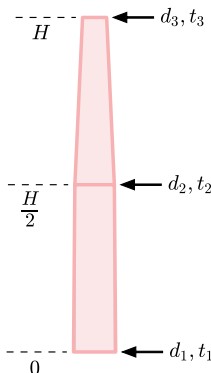

**Figure 3.** The parameterized turbine tower definition. The tower diameter and shell thickness are defined at the bottom, midpoint, and top of the tower, with the values linearly interpolated in between.

It may be beneficial to do a binary optimization in which each turbine can change the turbine group to which it belongs, but this greatly increases the complexity of the optimization and makes it gradient-free. Gradient-free optimization is more computationally expensive, which severely limits the number of design variables we can include in the problem. To maintain the gradient-based optimization, we assigned each turbine to one of the groups before starting the optimization. Although the turbines could move throughout the wind farm, once assigned a turbine could not switch to the other group. In this study, we
only examined an equal weighting of turbines in each group, but additional benefit may come from optimally choosing the number of turbines in each group.



We ran several cases in which different design variables were included in the problem to allow comparison of their effects on COE. In all, the design variables we included were the position of each turbine $(x_i, y_i)$, the tower height of each group $(H_1, H_2)$, the rotor diameter of each group $(D_1, D_2)$, the rated power of each group $(R_1, R_2)$, the tower diameter of each group $(d_{1,j}, d_{2,j})$, and the tower shell thickness of each group $(t_{1,j}, t_{2,j})$. Index j refers location on the tower (j=1 is at the bottom, j=2

at the midpoint, j=3 at the top), meaning there are six total variables to define diameter (three for each height group), and six to define the tower shell thickness. The blade chord and twist distributions also varied during the optimization, however these were implicit design variables.

The turbine layout and structural constraints were previously formulated in our multiple-hub-height study (Stanley et al.). Because rotor diameter was a design variable, the turbine spacing constraint was slightly reformulated such that the distance

between any two turbines in the wind farm was greater than the sum of the two rotor diameters. The rotor diameter and the turbine rating were constrained by the lowest and highest values that were included in the RotorSE optimization, as discussed in Sect. 2. The lower limits were never active in these optimizations; however, some of the upper limits were active as will be seen in the Results section. The optimization can be expressed:

minimize     COE

w.r.t.       $x_i,\ y_i,\ H_{1,2},\ D_{1,2},\ R_{1,2},\ d_{(1,j)},\ d_{(2,j)},\ t_{(1,j)},\ t_{(2,j)}$

$i = 1, \ldots, n;\ j = 1,\ 2,\ 3$

subject to   boundary constraints

$$\sqrt{(x - x_i)^2 + (y - y_i)^2} \geq 2D_{\text{rotor}}$$

$$H_1 - \frac{D_1}{2}, H_2 - \frac{D_2}{2} \geq 10 \text{ m}$$

$$d_{(1,j),(2,j)} \leq 6.3 \text{ m}$$

$$d_{(1,top),(2,top)} \geq 3.87 \text{ m} \tag{6}$$

$$\frac{3\,\Omega}{1.1} \geq f_{1,2} \geq 1.1\,\Omega$$

shell buckling margins: max thrust $\leq 1$

shell buckling margins: survival load $\leq 1$

$$\frac{d_{(1,j)}}{t_{(1,j)}}, \frac{d_{(2,j)}}{t_{(2,j)}} \geq 120$$

$$46 \text{ m} < D_1, D_2 < 160 \text{ m}$$

$$500 \text{ kW} < R_1, R_2 < 10,000 \text{ kW}$$

Note that $i$ is the index defining the wind turbine, and $j$ is the index describing the location on the tower.

The gradients for this optimization were all analytic. We calculated the partial derivatives of each small section of the model and included each part in a framework called OpenMDAO, which calculated the gradients of the entire system (Gray et al.,



2010). The analytic gradients were significant because they were more accurate, converged to better solutions, and converged on the solution much faster that finite difference gradients. More importantly, they allowed us to solve much larger optimization problems than would have been possible without.

We optimized two different wind farms, each with several different wind shear exponents and turbine spacing multipliers as will be explained later in this section. The first wind farm was a fictional 32-turbine wind farm with a circular boundary, shown in Fig. 4. This wind farm was optimized with wind data from the city of Alturas, California, shown in Fig. 5. For lack of reliable data, the average wind speed, $V_{mean}$, from each direction of this wind rose was assumed to be eight meters per second, shown in Fig 6. The second wind farm was the Princess Amalia wind farm, a real farm off the coast of the Netherlands which has 60 wind turbines, is shown in Fig. 4. This wind farm was optimized with the wind direction and average directional speed data from the true Princess Amalia farm, shown in Figs. 5 and 6. The farm boundary for the Princess Amalia wind farm was the convex hull of the original Princess Amalia layout. The turbines in the Princess Amalia wind farm are Vestas 2 Megawatt wind turbines, which have a rotor diameter of 80 meters. Therefore, for both wind farms we used a baseline rotor diameter of 80 meters and a baseline power rating of 2 Megawatts. The baseline hub height used in this study was 100 meters.

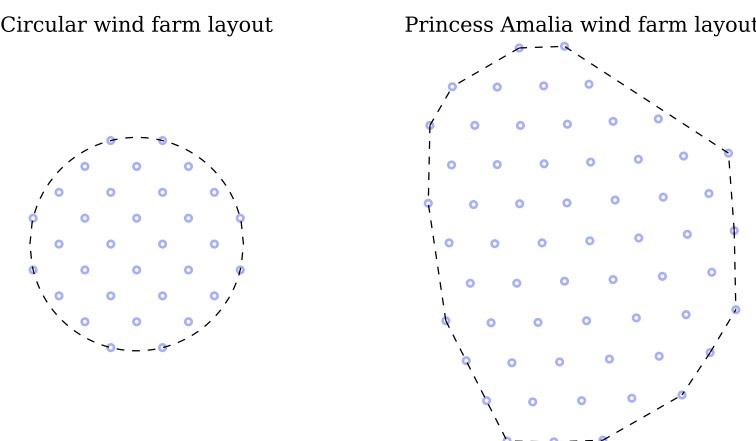

**Figure 4.** The two different wind farm designs that were optimized. On the left is a contrived circular wind farm design with 32 turbines. On the right is the Princess Amalia wind farm, an offset grid design with 60 wind turbines.

We optimized both of the wind farms shown in Fig. 4 with three different wind shear exponents (0.075, 0.175, 0.275), and three different spacing multipliers (0.5, 1.0, 1.5). The wind shear exponent defines how fast the wind speed changes with height, as seen in Equation 2. Low shear exponents are typical over open water or flat plains, while higher shear exponents exist in areas with obstructions, such as large trees or buildings. Figure 7 shows the wind speed profiles of the three shear exponents we used. For a shear exponent of 0.075, there is only an 8.6% increase in the wind speed from the reference height of 50 meters to 150 meters. For a shear exponent of 0.175 there is a wind speed increase of 21.2% for the same height difference, and for a shear exponent of 0.275 the wind speed increase is 35.3% from 50 to 150 meters. We also optimized each wind farm for different turbine spacings by adjusting the turbine locations by some spacing multiplier, $\beta$, which is simply some constant



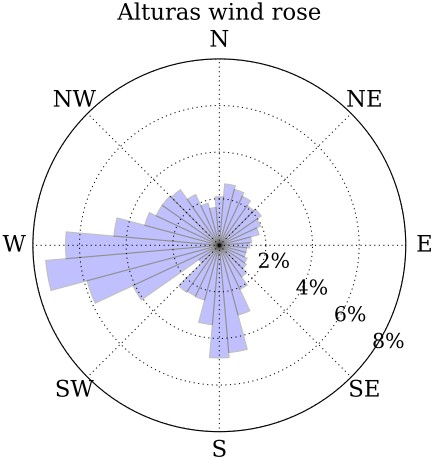
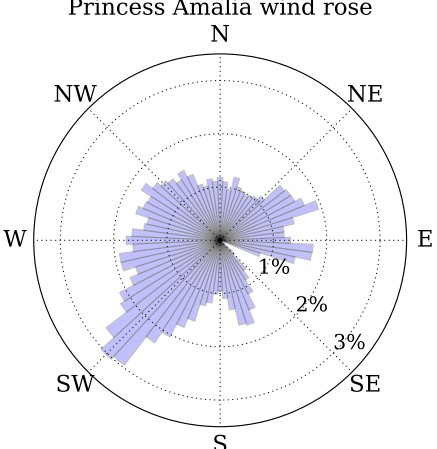

**Figure 5.** On the left, the wind direction distribution in Alturas, California, separated into 36 bins, every 10 degrees. On the right, the wind direction distribution of the Princess Amalia Wind Farm, separated into 72 bins, every 5 degrees.

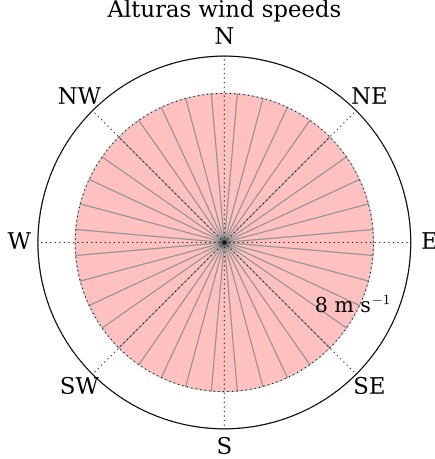
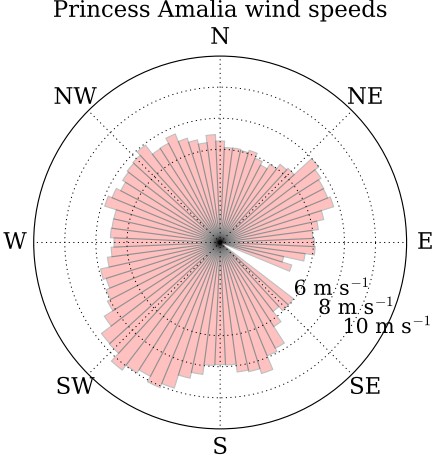

**Figure 6.** On the left, the assumed directionally averaged wind speeds for Alturas, California, separated into 36 bins, every 10 degrees. Each direction is assumed to have an average wind speed of 8 meters per second. On the right, the directionally averaged wind speeds of the Princess Amalia Wind Farm, separated into 72 bins, every 5 degrees.





multiplied to each turbine location. The wind farm boundaries were adjusted accordingly with the spacing multipliers, meaning the radius of the circular wind farm was also multiplied by the spacing multiplier, and the convex hull of the Princess Amalia farm was applied to the baseline turbine locations adjusted by the spacing multiplier. Figure 8 shows both of the wind farms adjusted by the spacing multipliers, as well as the turbine spacing in baseline rotor diameters. As the turbine designs were

optimized, these spacings (in rotor diameters) increased or decreased according to the new rotor diameters. Note that in the circular wind farm, the turbine distances are presented in the rows closely inline with the dominant wind direction (ten degrees south of west, see Fig. 5). The closest neighboring turbines are actually $\sqrt{2}/2$ multiplied by this value.

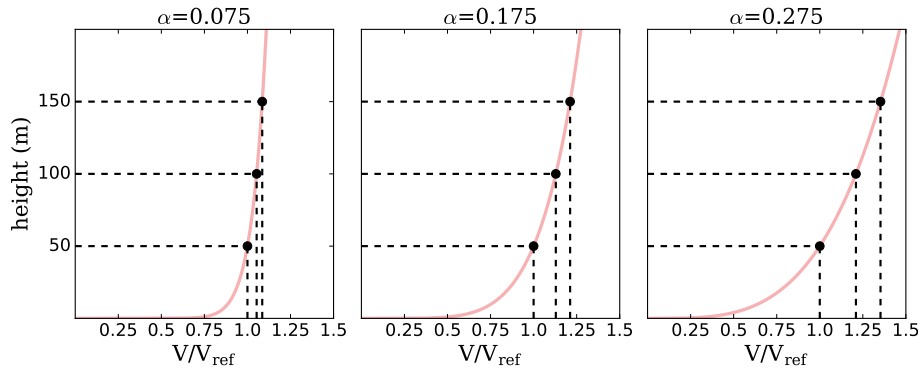

**Figure 7.** The wind speed profiles for various wind shear exponents. With lower shear exponents the wind speed does not vary dramatically with height. For higher wind shear there is a significant wind speed increase with height.

The results of gradient-based optimization, especially for problems with many local minima, are sensitive to the starting location. As in most optimization problems, there is no guarantee that the solution is the global solution. The best results can

be achieved with a multiple-start approach, where several different starting points are used for each condition, and the best solution is used. In our study, we ran fifty to hundreds of starting locations for each optimization case. For every optimization, we started each turbine location from the Princess Amalia or circular wind farm baseline locations in Fig. 8, each perturbed by a random amount. All of the other design variables were initialized randomly for each optimization.

## 3    Results

In this section we will discuss the optimization results of both wind farms, the apparent benefit of coupled turbine layout and design optimization, as well as the benefit of heterogeneous turbine design in a wind farm. We first present results from the 32-turbine circular wind farm optimizations and then compare to the 60-turbine Princess Amalia wind farm optimizations.




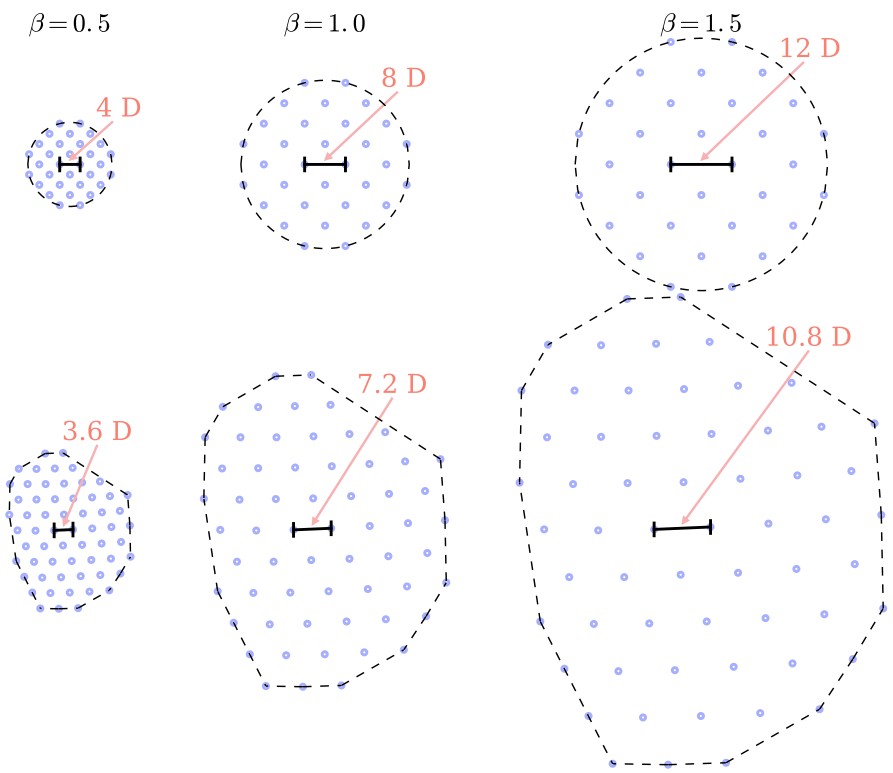

**Figure 8.** The six wind farm boundaries and associated baseline layouts optimized in this study. The same two layouts were multiplied by a spacing multiplier, $\beta = 0.5, 1.0, 1.5$, which changed the wind farm size and the averaging spacing between wind turbines. On the top is the 32-turbine, circular wind farm, and on the bottom is the 60-turbine, Princess Amalia wind farm. The turbine spacings, in baseline rotor diameters, are also displayed for each spacing multiplier in this figure.

### 3.1 Circular Wind Farm

Figure 9 shows the optimal COE results for the circular wind farm. As shown in the legends, the white points represent a layout-only optimization with the baseline turbine design, the gray indicate a sequential-turbine-design-then-layout optimization, the black squares show a coupled-turbine-design-and-layout optimization, and the half blue and pink points represent a coupled-design-and-layout optimization with two turbine groups. As expected, the general trends for all optimization runs show that the higher wind speed from high wind shear results in a lower, superior optimal COE. Additionally, the widely spaced wind turbines indicated by the larger spacing multipliers also result in lower COE due to less wake interaction between turbines. We will discuss each of these optimizations in detail below.





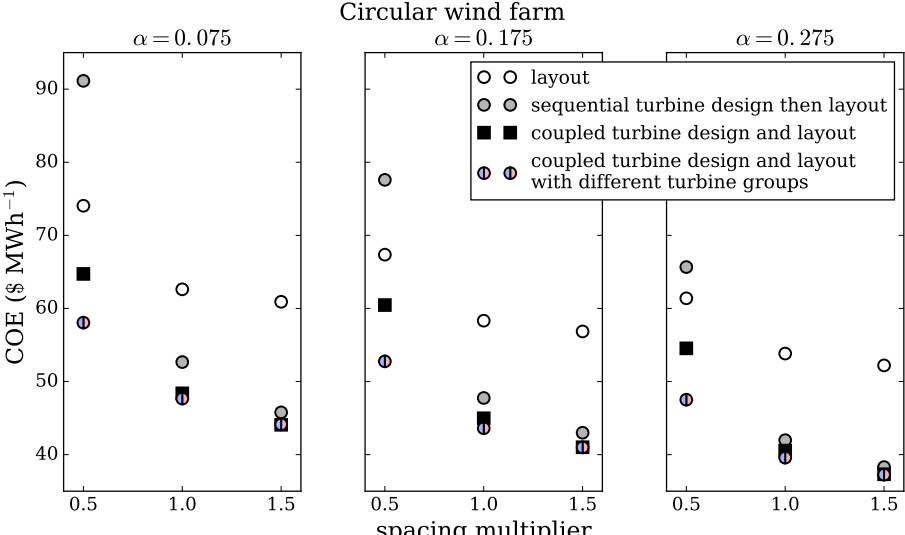

**Figure 9.** The optimal COE results for the circular wind farm layout with 32 turbines. Each of the subfigures corresponds to optimization runs with a different shear exponent, from left to right $\alpha = 0.075, 0.175, 0.275$. Within each subfigure, the x axis shows the size of the wind farm based on the spacing multiplier, from left to right $\beta = 0.5, 1.0, 1.5$. The different points represent the layout optimization with the baseline turbine design, sequential-turbine-design-then-layout optimization, coupled-layout-and-turbine-design optimization with homogeneous turbine design throughout the farm, and layout-and-turbine-design optimization with two different turbine design groups.

### 3.1.1 Circular Wind Farm: Sequential-Turbine-Design-then-Layout Optimization

The gray dots in Fig. 9 show the optimal COE results for a sequential optimization. First, a turbine was designed for minimal COE in isolation with the free stream wind conditions. This turbine design was then used in a wind farm where the layout was subsequently optimized. The rotor diameter was constrained such that the turbine spacing constraints would be satisfied in the baseline farm where the turbine would be installed. This was only applicable for the smallest wind farms, where $\beta = 0.5$. For each shear exponent, the optimal turbine design was the maximum rotor diameter and turbine rating allowed by the optimizer. The rotor diameter was constrained by the spacing constraint for $\beta = 0.5$, and by the bound constraint for other turbine spacings. Figure 10 shows the optimal isolated turbine designs for each shear exponent and spacing multiplier, as well as the baseline turbine design. Because these turbines are optimized in isolation and the spacing constraint was not active, the designs for $\beta = 1.0, 1.5$ are the same. When these optimized turbine designs are used in each wind farm instead of the baseline turbine design, there is a large COE improvement for the spacing multipliers of $\beta = 1.0, 1.5$. For $\beta = 1.0$, COE decreases 15.9–22.0% compared to an optimized wind farm with the baseline turbine design. For $\beta = 1.5$ the COE decrease is even larger, 24.8–26.6% across all shear exponents. For the smallest wind farm, $\beta = 0.5$, the turbine design optimized in isolation results in an extremely inefficient wind farm. When in the wind farm environment, exposed to much lower average wind speeds, this design results in a COE that is much worse than the baseline turbine design. The expense from a bigger and taller turbine,



coupled with the strong wake interactions among turbines that are so closely spaced means that for this wind farm, optimizing the turbine in isolation actually decreases the wind farm performance.

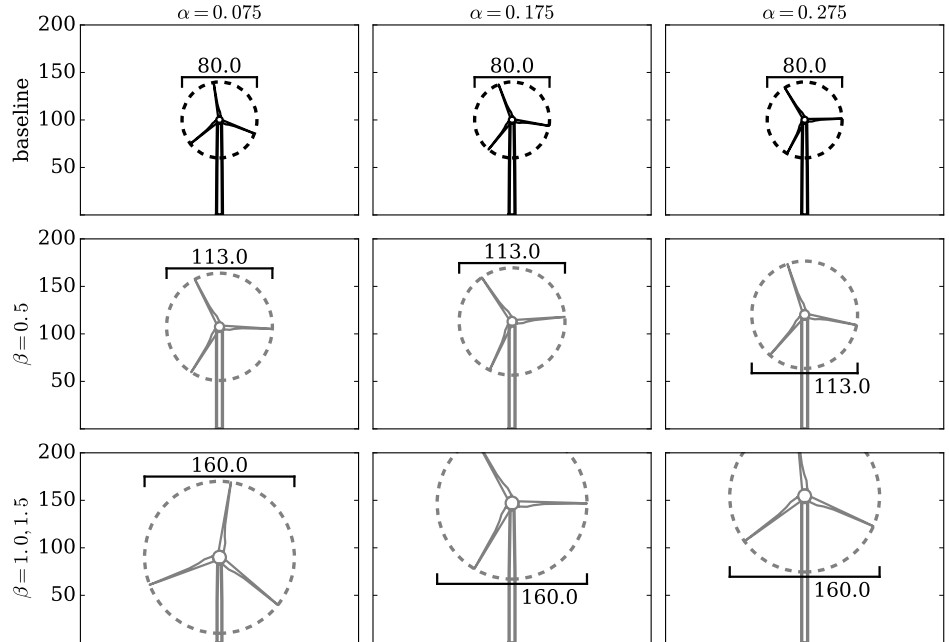

**Figure 10.** The optimal turbine heights and rotor diameters for the isolated turbine design optimization for the circular farm wind conditions. These designs were then used in the sequential turbine-design-then-layout optimizations. The columns, from left to right, show the turbines optimized for $\alpha = 0.075, 0.175$, and $0.275$. The rows, from top to bottom, show the baseline turbine design, the turbine optimized for the small wind farm ($\beta = 0.5$), and the turbine designs for the larger wind farms ($\beta = 1.0, 1.5$)

### 3.1.2 Circular Wind Farm: Coupled-Turbine-Design-and-Layout Optimization

Next we will discuss the optimization results of the coupled turbine-design-and-layout optimizations, represented by the black squares in Fig. 9. For every shear exponent and spacing multiplier, there is a large benefit to performing the coupled turbine-design-and-layout optimization compared to the layout-only optimization with the baseline turbine design. Additionally, and more importantly, the coupled optimization results in appreciably lower COE than the sequential design-then-layout optimization. Obviously for a spacing multiplier of $\beta = 0.5$, the coupled optimization is far superior to the sequential simply by being better than the baseline turbine design. For the spacing multiplier of $\beta = 1.0$, compared to the sequential optimizations, coupled optimization results in an additional 6.82%, 4.75%, and 2.65% COE improvement from layout only optimization for shear exponents $\alpha = 0.075, 0.175$, and $0.275$, respectively. For the largest wind farm, $\beta = 1.5$, the coupled optimization results in an additional 2.78%, 3.50%, and 1.88% COE improvement compared to the sequential case.

There are several conclusions we can make from both the sequential and coupled turbine design and layout optimizations. First, and most apparent, optimizing turbine design results in a much better wind farm than a farm in which the turbines are



selected arbitrarily or a priori. Second, and more importantly, optimizing turbine design coupled with the turbine layout is significantly better than optimizing the turbine design for the free stream wind conditions alone. In a wind farm, turbines rarely experience the free stream wind conditions as they are often waked by the other turbines in the farm. Therefore, the optimal turbine design is based on on average slower wind speeds than the free stream wind. This results in turbines with smaller hub

heights, rotor diameters, and rated powers. One could conceivably optimize the turbine design for some wind speed slower than the free stream and closer to the average speed in the wind farm, which would likely be better than optimizing the turbine design for the free stream wind speed. However, the average wind speed in a farm is dependent on the turbine layout, making it difficult to choose the correct speed for which to design the turbines. Thus, is important to couple the turbine design and layout optimization for a superior wind farm.

Figure 11 shows the optimal rotor diameters and hub heights for the coupled turbine design and layout optimizations. For a spacing multiplier $\beta = 0.5$, the turbines are very close together and in general are heavily waked. Thus to satisfy spacing constraints and because the average wind speed is very low, the optimal rotor diameter is small: about 90 meters. When the turbines are spaced farther apart, shown for the larger spacing multipliers, the optimal rotor diameter is much larger: closer to 120–130 meters. In these farms, wake interactions are not as severe, meaning that the extra power production from larger

rotors is worth the extra turbine capital cost. Also notice the trend of the optimal turbine height with wind shear exponent; for a low wind shear exponent, $\alpha = 0.075$, the wind speed does not drastically change with height (see Fig. 7). Therefore, for this wind condition it is desirable to have short hub heights with a lower turbine capital cost. For the higher shear exponents, $\alpha = 0.175, 0.275$, the wind speed increases much more with height (See Fig. 7). In these cases, for every spacing multiplier, the extra cost of building the taller turbines is made up for in the additional power produced from the high wind speeds. Remember

that a larger rotor diameter reduces the relative spacing between turbines in the farm, as the original spacing was based on a diameter of 80 meters.

In Fig. 12, the black points show the optimal rated powers for the turbines in each optimization case. The optimal rated power scales with the turbine rotor diameter and hub height. Higher turbine rating is expensive, therefore the small rotors and short turbines, which are more heavily waked and don't produce as much power, do not require a large power rating. The extra

cost is not justified by a very slight increase in power. For the high shear exponents and spacing multipliers, the turbines are exposed to faster wind speeds. These turbines are bigger and taller, and the extra power production from raising the rated power is worth the additional cost.

### 3.1.3   Circular Wind Farm: Coupled Turbine Design and Layout Optimization with Two Turbine Groups

Now we will discuss the most interesting case, the coupled-turbine-design-and-layout optimization with two different turbine

groups. The optimal COE results of these optimizations are shown with the blue and pink points in Fig. 9. Most visibly, for the smallest spacing multiplier, $\beta = 0.5$, there is a large COE improvement for the heterogeneous turbine design optimizations compared to the farms with homogeneous turbine design (shown by the black squares in Fig. 9). For this spacing multiplier, the heterogeneous turbine design farms reduce COE by 21.6%, 21.67%, and 22.6% compared to the layout-only optimization for shear exponents of $\alpha = 0.075, 0.175$, and $0.275$, respectively. The coupled optimizations with one turbine group reduce COE




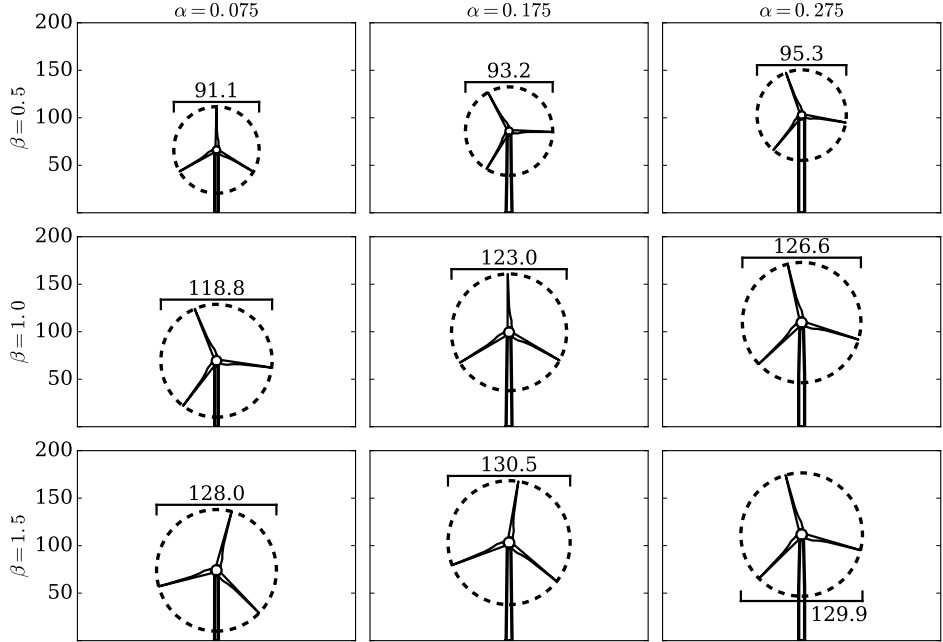

**Figure 11.** The optimal turbine heights and rotor diameters for the optimization runs with coupled layout and turbine design with homogeneous turbine design throughout the circular wind farm. Each column shows a different shear exponent, with $\alpha = 0.075, 0.175, 0.275$ from left to right. Each row shows a different farm spacing multiplier, with $\beta = 0.5, 1.0, 1.5$ from top to bottom.

by 12.59%, 10.24%, and 11.15%. For the smallest spacing multiplier, optimizing turbine design and layout with two turbine groups reduces COE by an additional 9–11.45% compared to just one turbine group. For the spacing multiplier $\beta = 1.0$, the coupled optimization with two turbine groups results in an additional 1.16–2.35% COE decrease compared to with one turbine group. This is much smaller than the more tightly packed wind farms, but still non-negligible. For the spacing multiplier

$\beta = 1.5$, the optimization with two turbine groups results in only an additional 0–0.12% COE decrease, indicating that when the turbines are spread very far apart there is no benefit to allowing multiple turbine designs in the same farm.

    The two different rotor designs in the same wind farm help to improve COE by reducing the wake interaction between wind turbines. By combining tall and short turbines, with large and small rotor sizes, there are more dimensions that the optimizer can manipulate to avoid wakes and improve performance. For the tightly packed wind farms, the turbine layout is greatly

limited by the turbine spacing constraints. Additionally, as the turbines are closer together, the wakes greatly reduce the wind speed as they have not had an opportunity to mix with the free steam air. Both of these factors mean there is a large benefit to avoiding the wakes of other turbines by any means possible. For the larger wind farms where the turbines are spaced farther apart, the wakes are not as detrimental and there is more area in which to avoid wakes in the horizontal plane without needing to change hub height or rotor diameter. In these cases, the heterogeneous turbine designs are not as beneficial.

Figure 13 shows the optimal rotor diameter and hub height of each turbine group for these cases of coupled turbine design and layout optimization with two different groups. For the spacing multiplier $\beta = 0.5$, when the turbines are very close together,





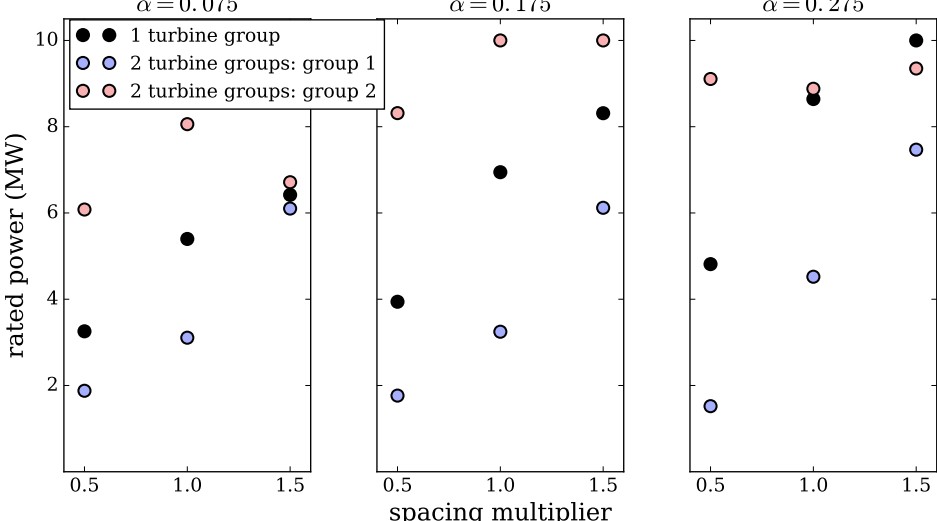

**Figure 12.** The optimal rated powers for the circular wind farm for the optimization runs with coupled layout and turbine design for both uniform wind farm turbine design and with two different turbine design groups. The three subfigures show a different shear exponent, with $\alpha = 0.075, 0.175, 0.275$ from left to right. within each subfigure, the x axis shows different farm spacing multipliers, with $\beta = 0.5, 1.0, 1.5$ from left to right.

there is a large difference in both the rotor diameter and hub height of each turbine group. Group 1 is extremely small and short, smaller than even the baseline rotor diameter, while group 2 is much larger. Even if turbines from each group were immediately adjacent to each other, there would be minimal wake interaction between the turbines. For the small wind farms, the sacrifice in power that comes from one very small and short turbine is made up for in the decreased wake interference between turbine

groups. Essentially, having two different turbine groups doubles the effective spacing between turbines, because turbines in different groups do not affect each other. For a larger spacing multiplier of $\beta = 1.0$, each turbine group is still remarkably different in size and height. The turbines are larger than they were for the smallest wind farm because the average wind speed is faster when the turbines are spread farther apart. Notice that, compared to the optimized turbines for $\beta = 0.5$, the smaller turbines when $\beta = 1.0$ are larger and overlap more with the taller, bigger turbines. In this case, the power increase from bigger

rotor diameters outweighs the benefit gained from reducing wake interference.

The turbine sizes for the largest wind farm, $\beta = 1.5$, demonstrate the multi-modality of the wind farm optimization problem. For this spacing multiplier, each turbine group is more similar than in the previous wind farm sizes. For the lowest shear exponent, $\alpha = 0.075$, both turbine groups are almost identical. For $\alpha = 0.175, 0.275$, there is some difference in each rotor diameter and hub height, although the difference is not as pronounced as it was for the smaller wind farms. However, Fig. 9

shows that for $\beta = 1.5$ the optimal COE from coupled turbine design and layout optimization is the almost exactly the same with one and two turbine groups. So, a wind farm with the homogeneous turbine design shown in the bottom row of Fig. 11,





and a wind farm with two different turbine designs shown in the bottom row of Fig. 13 result in a very similar optimal COE. The same optimal result is achieved with drastically different farms, each with different turbines and layouts.

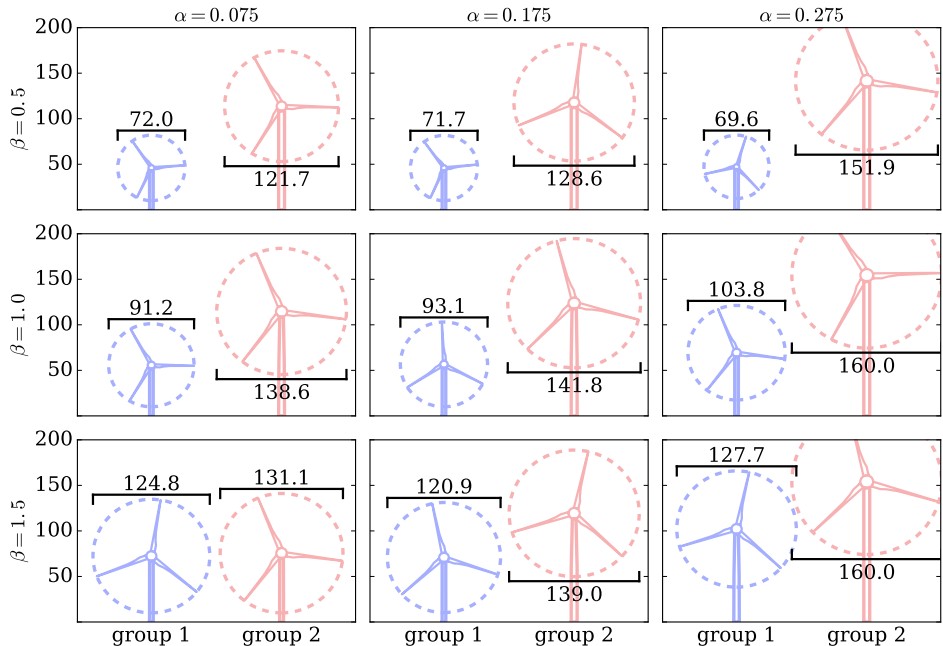

**Figure 13.** The optimal turbine heights and rotor diameters for the optimization runs with coupled layout and turbine design with two different turbine design groups for the circular wind farm. Each column shows a different shear exponent, with $\alpha = 0.075, 0.175, 0.275$ from left to right. Each row shows a different farm spacing multiplier, with $\beta = 0.5, 1.0, 1.5$ from top to bottom.

Figure 12 shows the optimal rated power of each height group for the optimization cases with two different turbine groups. The blue and pink dots in this plot correspond to the turbines of the same color in Fig. 13. As with the homogeneous turbine wind farm, the optimal rated power scales with the optimal turbine height and diameter. These larger, taller turbines are optimal in wind farms where they will be exposed to high wind speeds and produce large amounts of power. From a power production standpoint, it is undesirable to ever have a turbine's power limited by the rating. However, turbines with high ratings are more expensive, and not worth the cost if the turbine is generally producing low amounts of power. Therefore, the short, small turbines are optimal with a low, cheap power rating. The larger, taller turbines which produce much more electricity utilize the higher ratings.

## 3.2 Princess Amalia Wind Farm Results

In this section, we will discuss the results from the Princess Amalia wind farm optimizations. All of the optimizations that were performed with the circular, 32-turbine wind farm were repeated for the larger, 60-turbine Princess Amalia wind farm.



We will show and briefly discuss the optimal COE results; however, the optimal turbine designs for the Princess Amalia wind farm optimizations were very similar to those for the circular wind farm and therefore will not be included in this paper.

Figure 14 shows the COE results for the 60-turbine Princess Amalia wind farm optimizations. The trends are similar to the smaller, circular wind farm. Coupled turbine design and layout optimization is superior to optimizing each sequentially,

especially for the smaller wind farms where the wind speeds are much lower than the free stream. For the farms with closely spaced wind turbines, two different turbine designs in the same farm are significantly better than the farms optimized with a homogeneous turbine design. If the largest wind farms ($\beta = 1.5$) benefit from two different turbine design groups, that benefit is negligible. The optimal COE values for the Princess Amalia wind farm are slightly lower across the board than the circular wind farm COE values. This is partly because there are more turbines in the Princess Amalia wind farm so a smaller portion

of the total cost comes from overhead but also is partly due to the Princess Amalia wind turbines being spaced slightly farther apart than those in the circular wind farms. Another major difference between the optimal COE values of each wind farm is in the optimization case with two turbine design groups. For the Princess Amalia wind farm and a spacing multiplier of 0.5, two turbine groups provides and additional COE decrease of 6.13–9.11% compared to the wind farm with homogeneous turbine design. This is significant; however, it is not as large as the 9.01–11.45% additional COE decrease in the circular wind farm

optimizations for the same spacing multiplier. Again, the main cause of this seems to be that the turbines in the circular wind farms are slightly closer together than the turbines in the Princess Amalia wind farms.

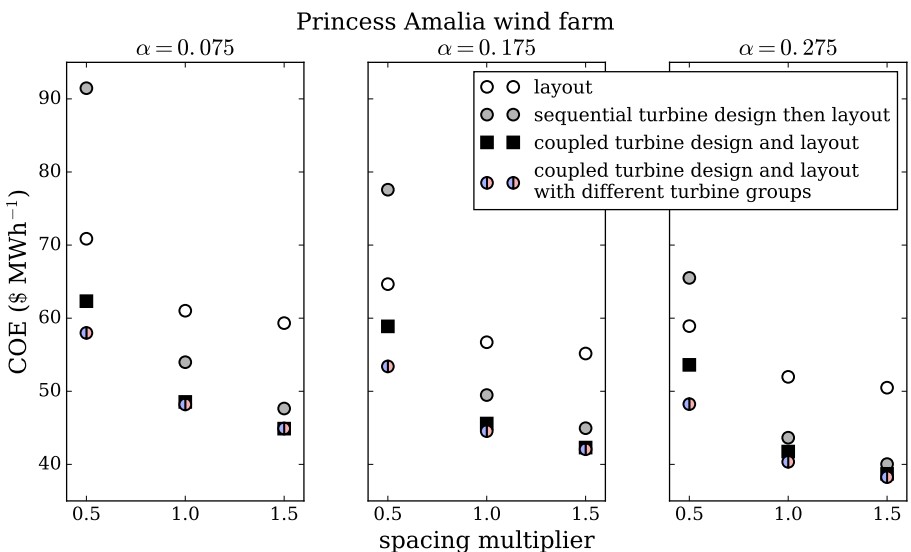

**Figure 14.** The optimal COE results for the Princess Amalia wind farm layout with 60 turbines. Each of the subfigures corresponds to optimization runs with a different shear exponent, from left to right $\alpha = 0.075, 0.175, 0.275$. Within each subfigure, the x axis shows the size of the wind farm based on the spacing multiplier, from left to right $\beta = 0.5, 1.0, 1.5$. The different points represent the layout optimization, sequential turbine-design-then-layout optimization, coupled layout-and-turbine-design optimization with homogeneous turbine design throughout the farm, and layout-and-turbine-design optimization with two different turbine design groups.



**Table 1.** The percent COE decrease of the various optimization cases with respect to layout-only optimization. This table does not show the overall desirability of the optimal wind farm, but the relative improvement of different considerations of turbine design optimization. In the table are shown results for each shear exponent, $\alpha$, as well as each spacing multiplier, $\beta$, in which the smaller spacing multipliers represent farms with turbines that are more closely spaced.

**Percent COE decrease compared to layout only optimization**

**circular wind farm**

| optimization case | $\alpha = 0.075$ | | | $\alpha = 0.175$ | | | $\alpha = 0.275$ | | |
| --- | --- | --- | --- | --- | --- | --- | --- | --- | --- |
| | $\beta{=}0.5$ | $\beta{=}1.0$ | $\beta{=}1.5$ | $\beta{=}0.5$ | $\beta{=}1.0$ | $\beta{=}1.5$ | $\beta{=}0.5$ | $\beta{=}1.0$ | $\beta{=}1.5$ |
| sequential | -23.07 | 15.90 | 24.84 | -15.19 | 18.13 | 24.37 | -6.97 | 22.01 | 26.64 |
| coupled: 1 group | 12.59 | 22.72 | 27.62 | 10.24 | 22.88 | 27.87 | 11.15 | 24.66 | 28.52 |
| coupled: 2 groups | 21.60 | 23.88 | 27.54 | 21.67 | 25.23 | 27.90 | 22.60 | 26.46 | 28.64 |

**Princess Amalia wind farm**

| optimization case | $\alpha = 0.075$ | | | $\alpha = 0.175$ | | | $\alpha = 0.275$ | | |
| --- | --- | --- | --- | --- | --- | --- | --- | --- | --- |
| | $\beta{=}0.5$ | $\beta{=}1.0$ | $\beta{=}1.5$ | $\beta{=}0.5$ | $\beta{=}1.0$ | $\beta{=}1.5$ | $\beta{=}0.5$ | $\beta{=}1.0$ | $\beta{=}1.5$ |
| sequential | -29.06 | 11.54 | 19.70 | -19.98 | 12.74 | 18.52 | -11.19 | 16.02 | 20.70 |
| coupled: 1 group | 12.05 | 20.45 | 24.34 | 8.94 | 19.61 | 23.32 | 9.00 | 19.66 | 23.33 |
| coupled: 2 groups | 18.18 | 21.01 | 24.30 | 17.41 | 21.45 | 23.74 | 18.11 | 22.37 | 24.24 |

Table 1 shows how the optimal COE results for each wind farm compared to the layout optimization with the baseline wind turbine design. These numbers compare the relative benefit of performing turbine design with the various scenarios mentioned. High numbers represent a large COE decrease compared to the layout-only optimization for a given shear exponent and spacing multiplier combination; they do not necessarily represent a low COE. There are a few interesting numbers in this table. Most obviously is the negative values (shown in red) for the sequential optimization with a spacing multiplier of 0.5. For these farms, sequential optimization is actually worse than the baseline. Also notice the COE decrease from coupled optimization with one group to two groups. For $\beta = 0.5$, there is a huge benefit to having two groups, for $\beta = 1.0$ there is a small benefit, and for $\beta = 1.5$ there is no benefit at all. Finally, the benefit of coupled optimization with one group compared to sequential optimization is important. Again, there is a huge benefit to coupled optimization for the smallest spacing multiplier, and this relative benefit decreases as the wind farm size grows. However, even for $\beta = 1.5$, there is an appreciable benefit to coupled design-and-layout optimization compared to sequential.

## 4 Conclusions

The purpose of this study was to optimize wind turbine design and turbine layout in various wind farms. There was a particular focus on benefits from coupled turbine design and layout optimization, as well as having different turbine designs in





the same wind farm. We simulated wind farms in this study by modifying and combining a variety of separate wind farm models, including the FLORIS wake model, portions of TowerSE and Plant_CostsSE, and a surrogate of RotorSE. Wind farms were optimized to minimize COE using turbine layout and turbine design including hub height, rotor diameter, rated power, tower diameter, and tower shell thickness as design variables, as well as blade chord and twist distributions as implicit design variables. We optimized two wind farms, a contrived 32-turbine circular wind farm, and the 60-turbine Princess Amalia wind farm. Both were optimized for a range of shear exponents and turbine spacings.

Our main conclusions are twofold: coupled turbine design and layout optimization provides significant benefits compared to optimizing sequentially, and for many wind farms, two different turbine designs can greatly reduce the cost of energy. Without exception, coupled design and layout optimization performed better than optimizing the turbine design followed by the turbine layout. For a turbine design optimized in isolation, as was done in the sequential case, it was always most optimal to have a rotor diameter as large as the constraints would allow. Also, in coupled wind farm optimization, the wind farms with large spacing multipliers tended towards large rotor diameters. For this reason, the smallest wind farms benefited most from the coupled design and layout optimization, because the wind speeds were slow from strong wake interactions, and optimal rotor diameter was small—much different than the turbines optimized in isolation. The coupled optimization was better than sequential optimization, regardless of the wind shear exponent.

Including two different turbine designs in the same wind farm can be very beneficial in reducing wake interference between wind turbines and result in a lower COE compared to a farm with all identical wind turbines. For wind turbines that are close together, wake interactions are very strong between turbines. With different turbine sizes, the hub height and rotor diameter can be optimized along with layout to avoid wakes in the vertical plane along with the horizontal plane. For a spacing multiplier $\beta = 0.5$, indicating very closely spaced wind turbines, our optimization results show that two different turbine designs can reduce COE by an additional 10% compared to wind farms with homogeneous turbine design. For $\beta = 1.0$, the farms with heterogeneous turbine designs are marginally better than the optimized farms with uniform design by 1–3%. For the largest farms, $\beta = 1.5$, there is no benefit to having two different turbine designs in the same wind farm. When the turbines are very far apart, the wake interactions are weak enough that the turbines can approach the turbine design optimized in isolation. Again, the two turbine groups was better than one group, regardless of the wind shear exponent. In our previous study, we optimized wind farms with two different turbine heights and keeping the rotor diameter and rated powers constant. We found that wind farms with low wind shear benefited much more from different hub heights than wind farms with higher wind shear (Stanley et al.). However, in this study the rotor diameters and rated powers were also optimized. The turbine design could be customized to best utilize the different wind resources available with different wind shear values. Therefore, the benefit from having two different turbine designs in the same wind farm is independent of the wind shear exponent.

*Code and data availability.* Forthcoming



*Competing interests.* The authors declare that they have no conflict of interest.

*Acknowledgements.* The authors developed this journal article based on funding from the Alliance for Sustainable Energy, LLC, Managing and Operating Contractor for the National Renewable Energy Laboratory for the U.S. Department of Energy.

Funding for the work was provided by the DOE Office of Energy Efficiency and Renewable Energy, Wind Energy Technologies Office.

5    The U.S. Government retains and the publisher, by accepting the article for publication, acknowledges that the U.S. Government retains a nonexclusive, paid-up, irrevocable, worldwide license to publish or reproduce the published form of this work, or allow others to do so, for U.S. Government purposes.





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
