# Peer review of "Coupled Wind Turbine Design and Layout Optimization with Non-Homogeneous Wind Turbines"

_Wind Energy Science, 2018_

## Referee Comment (RC1) · Anonymous Referee #1 · 6 Sep 2018

The paper presents an interesting analysis on the optimization of wind farm layout. The authors find that an integral optimization approach of the wind farm layout and turbine design together gives improved results when compared to more traditional approaches in which first the turbine design is optimized before considering the optimization of the wind farm layout. The results are interesting to the community and for publications on the Wind Energy Science journal. I believe that several aspects need some clarification as indicated in the list below.

**** General comments

– In the abstract the authors should mention under which conditions the 2 to 5% reduction was obtained (which spacing, wind shear, etc) and mention the benefit is larger when the inter turbine spacing is smaller.

– Mention in the abstract that only two turbine designs are allowed in the optimization procedure in a fixed one to one ratio. Now the abstract gives the impression the turbine design is changed for each turbine.

– In several places the authors mention that certain modeling choices were given by the need to restrict the computational time used by the optimization procedure.

– Page 3: What is the wake profile in the FLORIS model? The authors define the wake interaction model in detail, but do not provide any details on the actual wake model. Please also use L only for one thing.

– Page 7 cost model: The authors refer to a reference that is not yet available to the referee. Without access to this paper it is impossible to see what the authors exactly did. If the paper will appear soon there is no need to repeat things here if all is identical, but for the moment this cannot be checked.

– Page 7 section 2.6: The authors say it does not matter too much how many different groups are used, but no actual numbers are mentioned. Please provide these numbers to back up this statement.

– page 9 last section (also page 11 top): I found the description of the "spacing multiplier" very confusing. Figure 8 helps a lot, but the text in the mentioned paragraph should be clarified.

– page 14 first section: Did the authors try to see what happens when they use the turbine designs obtained a a training wind farm as input for the actual optimization algorithm. How close would the result be to the coupled optimization approach? Obviously the coupled optimization approach always gives a better results (assuming obviously that the model would be perfect), but how much is the difference? In the results we namely also see that for very small spacings the "sequential turbine design then layout" is worse than just using the "layout" optimization. Is this because the reference turbines are already somewhat optimized to be located in a wind farm? And would a better design of more appropriate reference turbines reduce the benefit of the coupled optimization approach?

– How exactly does this work compare to earlier work by the same authors, i.e.

Stanley, A. P. J., Thomas, J., Ning, A., Annoni, J., Dykes, K., and Fleming, P., "Gradient-Based Optimization of Wind Farms with Different Turbine Heights," Wind Energy Symposium, Grapevine, TX, AIAA, Jan. 2017. doi:10.2514/6.2017-1619

Stanley, A. P. J., Ning, A., and Dykes, K., "Benefits of Two Turbine Rotor Diameters and Hub Heights in the Same Wind Farm," Wind Energy Symposium, Kissimmee, FL, Jan. 2018. doi:10.2514/6.2018-2016

These papers are not referenced at the moment, but seem to follow for a big part the optimization philosophy outlined here, with the present work focusing on the coupled layout wind turbine design approach. It may be possible to compare the results obtained here with the previous results.

– Can the authors compare their optimization results with methods already presented in the literature?

– The authors only show results of the turbine designs obtained using the optimization. I would also be interested in seeing the corresponding wind farm layouts of the most optimal configuration that was found.

**** Specific comments

– Page 1: References on the mentioned gradient-free and gradient-based optimization methods are missing.

– Page 2: in this study for additional improvements. -> It should read "in the presented approach" as the authors do not actually consider yaw optimization in this study.

– Page 2-3: Please clarify the numbers presented here, i.e. the increases in production / reduction in the cost of energy are determined with respect to some reference case, but not for all cases the respective reference cases are clearly defined.

– Page 4: Rotation of each wind turbine -> rotation of the blades

– Page 4: Please define x in equation 5.

– Page 4: The direction data we had -> Please provide the appropriate references / explain. I assume most of the data in figure 5/6 is actually obtained from some literature reference.

– Page 5 line 20: Maybe I missed it, but please give the airfoil used in this work.

– Page 5 figure 1: Please provide units of Frequency shown here.

– Page 6 line 1: What 10 groups are used?

– Page 6 figure 2: Please give the appropriate units in the figure.

– Page 8 equation 6: What is D_rotor? In the optimization both D1 and D2 are used. Which one is used for D_rotor?

– Page 8 equation 6: Please write D(1,j),D(2,j), etc

– Page 8 equation 6: Shell buckling does not seem to be defined in the paper. Why are these values used? What are the units?

– Page 9 line 2: that -> than

– page 9 last section: the "shear exponent" does not seem to be properly defined yet.

– page 11 line 13: "random amount" -> How much approximately.

– page 12 figure 8: Please mention that D=80 meters (and fixed) for the purposes of this plot. This only became clear much later.

– page 20 table 1: For clarity mention in the caption what the reference case is.

---

## Referee Comment (RC2) · Anonymous Referee #2 · 6 Oct 2018

[referee-annotated manuscript omitted]

---

## Author Comment (AC1) · 30 Oct 2018

[12pt]report [margin=1in]geometry color graphicx float hyperref

[Figure]

**Response to Reviewer 1**

Andrew PJ Stanley and Andrew Ning

October 30, 2018

Thank you for your thorough review of the manuscript and for your comments! We will address each of your comments and questions individually.

Question/Comments are in black.

The corresponding responses are immediately below in blue.

**** General comments

In the abstract the authors should mention under which conditions the 2 to 5% reduction was obtained (which spacing, wind shear, etc) and mention the benefit is larger when the inter turbine spacing is smaller.

The abstract was changed to reflect this comment.

Mention in the abstract that only two turbine designs are allowed in the optimization procedure in a fixed one to one ratio. Now the abstract gives the impression the turbine design is changed for each turbine.

The abstract was changed to reflect this comment.

In several places the authors mention that certain modeling choices were given by the need to restrict the computational time used by the optimization procedure.

Yes, some modeling choices were made to restrict time. In an optimization framework where a model will be called several times every iteration, computationally expensive models become infeasible. For example, we created a surrogate model using RotorSE. Our surrogate model takes about 3.4E-5 seconds. With the surrogate model, a single function call analyzing one of the wind farms with two different turbine designs takes about 0.2 seconds, so the time to call the surrogate model is negligible. A single function call of RotorSE on the other hand takes about 0.85 seconds, so using RotorSE instead of the surrogate in the function call to analyze the wind farm would take about 1.9 seconds (the original 0.2 seconds plus 2*0.85 seconds for two different RotorSE calls). This is almost 10 times as long per function call! The optimizations that we ran could already take as long as 6-8 hours, so to multiply that by 10 would mean a single optimization could take several days. For this study, when we ran hundreds of random starts for many different cases, this was infeasible.

Page 3: What is the wake profile in the FLORIS model? The authors define the wake interaction model in detail, but do not provide any details on the actual wake model. Please also use L only for one thing.

An explanation was added about how the version of FLORIS that we used was modified to consider the 3-D flow field. All of the details of the FLORIS modeled were not re-presented here as they are detailed in the cited papers and are used exactly as defined in those papers Also, the equation was adjusted to clarify total loss and individual loss from each turbine.

Page 7 cost model: The authors refer to a reference that is not yet available to the

referee. Without access to this paper it is impossible to see what the authors exactly did. If the paper will appear soon there is no need to repeat things here if all is identical, but for the moment this cannot be checked.

Yes this is unfortunate. We expect this paper to be published soon, in the meantime we will attach a pre-print version of the paper!

Page 7 section 2.6: The authors say it does not matter too much how many different groups are used, but no actual numbers are mentioned. Please provide these numbers to back up this statement.

Relating to the last comment, the intention is that the reader would refer to the citation for the specifics on this topic. We will attach a pre-print version of the paper.

page 9 last section (also page 11 top): I found the description of the "spacing multiplier" very confusing. Figure 8 helps a lot, but the text in the mentioned paragraph should be clarified.

This portion was reworked for additional clarity.

page 14 first section: Did the authors try to see what happens when they use the turbine designs obtained a a training wind farm as input for the actual optimization algorithm. How close would the result be to the coupled optimization approach? Obviously the coupled optimization approach always gives a better results (assuming obviously that the model would be perfect), but how much is the difference? In the results we namely also see that for very small spacings the "sequential turbine design then layout" is worse than just using the "layout" optimization. Is this because the reference turbines are already somewhat optimized to be located in a wind farm? And would a better design of more appropriate reference turbines reduce the benefit of the coupled optimization approach?

Excellent point! We added a paragraph to the end of the conclusion that includes a recommendation for future research to include the consideration of a "training farm" for sequential optimization. As for the sequential optimization being worse for very small spacings, yes that is correct. When the turbine is optimized in isolation, it is optimal to be large because it is always exposed to the high free stream wind speeds. In a farm, especially the small farm, the wind speeds that the turbine actually is exposed to are much lower from wake interference, so in these cases it is more optimal to be smaller. It happens that the baseline design is close to the optimal turbine for the closely spaced wind farm, which makes it more optimal than the larger sequential optimization.

How exactly does this work compare to earlier work by the same authors, i.e. Stanley, A. P. J., Thomas, J., Ning, A., Annoni, J., Dykes, K., and Fleming, P., "GradientBased Optimization of Wind Farms with Different Turbine Heights," Wind Energy Symposium, Grapevine, TX, AIAA, Jan. 2017. doi:10.2514/6.2017-1619 Stanley, A. P. J., Ning, A., and Dykes, K., "Benefits of Two Turbine Rotor Diameters and Hub Heights in the Same Wind Farm," Wind Energy Symposium, Kissimmee, FL, Jan. 2018. doi:10.2514/6.2018-2016 These papers are not referenced at the moment, but seem to follow for a big part the optimization philosophy outlined here, with the present work focusing on the coupled layout wind turbine design approach. It may be possible to compare the results obtained here with the previous results.

These papers were added and explained in the literature review! This paper is the culmination of these two conference papers and the referenced journal submission.

Can the authors compare their optimization results with methods already presented in the literature?

A brief sentence on how our results compared to that of the literature was added to the last paragraph of the introduction. This addition along with the rest of that paragraph compare the similarity of results as well as the key differences between our paper and

those done by others.

The authors only show results of the turbine designs obtained using the optimization. I would also be interested in seeing the corresponding wind farm layouts of the most optimal configuration that was found.

We considered this at some length before the original submission, and decided to leave out the optimal wind farm layouts simply due to the shear number of figures it would require. For each wind farm, there are 3 spacing multipliers considered, and for each spacing multiplier there are 3 shear exponents. Then there are 4 optimizations that were run (layout, sequential, coupled with one group, coupled with two groups). That is a total of 72 layout plots to include all of the relevant results. To include one or two layouts would not be interesting, because the important part is the comparison between each case, we thus decided to leave them out.

Although they won't fit in this paper, we have posted the figures and added this link to the paper at the top of the Results section:

https://github.com/byuflowlab/stanley2018-turbine-design/tree/master/latex-files/ Figures/optimalLayouts

**** Specific comments

Page 1: References on the mentioned gradient-free and gradient-based optimization methods are missing.

We double checked the references and they appear to be listed (lines 9-11 on pg 2).

Page 2: in this study for additional improvements. -> It should read "in the presented approach" as the authors do not actually consider yaw optimization in this study.

This change was made in the updated document.

Page 2-3: Please clarify the numbers presented here, i.e. the increases in production / reduction in the cost of energy are determined with respect to some reference case, but not for all cases the respective reference cases are clearly defined.

The missing reference case was added.

Page 4: Rotation of each wind turbine -> rotation of the blades

This change was made in the updated document.

Page 4: Please define x in equation 5.

"x" was changed to "V" to represent wind speed in the updated document.

Page 4: The direction data we had -> Please provide the appropriate references / explain. I assume most of the data in figure 5/6 is actually obtained from some literature reference.

Appropriate references were added in the updated document

Page 5 line 20: Maybe I missed it, but please give the airfoil used in this work.

The source with the airfoils used was added in the updated document.

Page 5 figure 1: Please provide units of Frequency shown here.

Frequency refers to the probability and is unit-less.

Page 6 line 1: What 10 groups are used?

The groups were randomly selected as is typical in k-fold cross-validation.

Page 6 figure 2: Please give the appropriate units in the figure.

The axes have been normalized and are thus unit-less.

Page 8 equation 6: What is D_rotor? In the optimization both D1 and D2 are used. Which one is used for D_rotor?

This line was replaced with "spacing constraints" for clarity.

Page 8 equation 6: Please write D(1,j),D(2,j), etc

"D" refers to the rotor diameter, where "j" is the index referring to the location up the height of the tower. Thus the index is inappropriate for this variable.

Page 8 equation 6: Shell buckling does not seem to be defined in the paper. Why are these values used? What are the units?

This was clarified on page 8, and an appropriate equation was added to explain the shell buckling margins.

Page 9 line 2: that -> than

This change was made in the updated document.

page 9 last section: the "shear exponent" does not seem to be properly defined yet.

Shear exponent is defined in equation 2 on page 3.

page 11 line 13: "random amount" -> How much approximately.

[Figure]

This was clarified to be up to two baseline rotor diameters in the x and y coordinates.

page 12 figure 8: Please mention that D=80 meters (and fixed) for the purposes of this plot. This only became clear much later.

This was added to the figure and mentioned in the updated document.

page 20 table 1: For clarity mention in the caption what the reference case is.

The reference case is in the caption: "layout only optimization"

---

## Author Comment (AC2) · 30 Oct 2018

Referee 1 noted that we cited a paper that is currently in review, and is not available at the moment. We expect that this paper will be published soon, but for now we have attached a preprint version of this paper:

"Optimization of Turbine Design in Wind Farms with Multiple Hub Heights, Using Exact Analytic Gradients and Structural Constraints" by Andrew PJ Stanley, Andrew Ning, and Katherine Dykes

[Figure]

**Supplement:**

DOI: xxx/xxxx

RESEARCH ARTICLE

**Optimization of Turbine Design in Wind Farms with Multiple Hub Heights, Using Exact Analytic Gradients and Structural Constraints**

Andrew PJ Stanley*[1] | Andrew Ning[1] | Katherine Dykes[2]

[1]Department of Mechanical Engineering, Brigham Young University, Provo, Utah, USA

[2]National Wind Technology Center, National Renewable Energy Laboratory, Golden, Colorado, USA

**Correspondence**
*Andrew PJ Stanley, Department of Mechanical Engineering, 350 EB, Brigham Young University, Provo, UT 84602
Email: stanley_andrewpj@byu.net

**Abstract**

Wind farms are generally designed with turbines of all the same hub height. If wind farms were designed with turbines of different hub heights, wake interference between turbines could be reduced, lowering the cost of energy (COE). This paper demonstrates a method to optimize onshore wind farms with two different hub heights using exact, analytic gradients. Gradient-based optimization with exact gradients scales well with large problems and is therefore preferable in this application over gradient-free methods. Our model consisted of the following: a modified version of the FLOw Redirection and Induction in Steady State wake model that accommodated three-dimensional wakes and calculated annual energy production, a wind farm cost model, and a tower structural model, which provided constraints during optimization. Structural constraints were important to keep tower heights realistic, and account for the additional mass required from taller towers and higher wind speeds. We optimized several wind farms with tower height, diameter, and shell thickness as coupled design variables. Our results indicate that wind farms with small rotors, low wind shear, and closely spaced turbines can benefit from having two different hub heights. Specifically, a 9 by 9 grid wind farm with 70 meter rotor diameters and a wind shear exponent of 0.08 realized a 4.9% reduction in COE by using two different tower sizes. If the turbine spacing was reduced to 3 diameters, the reduction in COE decreased further to 11.2%. Allowing for more than two different turbine heights is only slightly more beneficial than two heights and is likely not worth the added complexity.

**KEYWORDS:**
gradient-based optimization, analytic gradients, structural constraints, different hub heights, FLORIS wake model, tower sizing

**NOMENCLATURE**

| | | |
|---|---|---|
| $AEP$ [1] | = | annual energy production (GWh) |
| $COE$ | = | cost of energy ($/MWh) |
| $\alpha$ | = | wind shear exponent |
| $x$ | = | wind turbine x location (m) |
| $y$ | = | wind turbine y location (m) |
| $z$ | = | wind turbine hub height (m) |
| $\vec{d}$ | = | tower diameter vector (m) |
| $\vec{t}$ | = | tower shell thickness vector (m) |
| $H$ | = | group hub height (m) |

**1 | INTRODUCTION**

As wind turbines extract energy from the air and convert it to power, an area of reduced wind speed, known as a wake, is formed behind each wind turbine. Because the air in a wake has less momentum, a wind turbine located in a wake cannot extract as much energy and therefore cannot produce as much power. Several solutions have been developed to help remedy this problem, including layout optimization of the wind farm[1,2,3] and rotor yaw control[4,5]. It is typical to design wind farms with a single turbine type, which has been designed before any farm layout and control optimization. However, more than one turbine type in a wind farm may help to further decrease negative wake effects. In this paper we will explore the possible benefits of optimizing farm design with different turbine hub heights in decreasing COE. Specifically, we will analyze the benefits of including turbines of two different hub heights in the same onshore wind farm.

Mixed turbine wind farms have been and are currently being studied to increase wind farm productivity. Graf et al. used a genetic algorithm to study layout optimization of a wind farm, while allowing the turbine type to change. They found that the optimal design had two turbine types, and was slightly better than a farm with only one turbine type[6]. Several studies have also already explored the use of different turbine heights in the same wind farm. Chen et al. used a genetic algorithm to optimize a wind farm layout of 25 turbines by changing the position and height of each turbine between two predefined heights. They found that the power increased by as much as 13.5% and the cost per unit of energy produced decreased 0.4%[7]. Hazra et al. used a particle swarm method to optimize a wind farm, in which the turbine height and rotor radius are both design variables. In a 10-turbine wind farm, they found a 12.8% reduction in the cost of power[8].

More recently, Chen et al. used a greedy algorithm to optimize a wind farm layout and hub heights. Using a particle wake model they found that for wind farms with complex terrain, COE could be decreased by more than 8% when optimizing with two different hub heights rather than just one. They also found marginal benefits using the Jensen wake model for wind farms on flat terrain[9]. Vasel-Be-Hagh and Archer also used a greedy algorithm in which they optimized a wind farm with only hub height as a design variable. They showed a 2% increase in power production when different hub heights are in the same farm. Additionally, Vasel-Be-Hagh and Archer explored situations with two turbines in line with the wind. They showed how the turbine spacing, rotor diameter, and wind shear all affect the power production of the two turbines. Low wind shear, small rotor diameter, and closely spaced turbines produced more power when using different hub heights[10].

Wind farms are large investments, and small improvements of less than 1% can greatly reduce costs and increase profitability over the life of the farm. The results of the studies discussed previously all show significant decreases in the COE of wind farms. All of the mentioned studies indicated that there are farms that can benefit from including turbines with heterogeneous designs—and different hub heights in particular.

Wind farm optimization can be a large problem with many potential design variables and constraints. Optimizing turbine layout introduces two design variables for each turbine. Including tower height introduces another variable per turbine. Other variables can also be added for yaw control, the rotor, tower diameter and thickness, and turbine type. For a wind farm with 25 wind turbines, the problem can easily have thousands of design variables. Gradient-based optimization with analytic gradients is effective for this number of design variables. The number of function calls required to converge using gradient-free optimization scales approximately quadratically with the number of design variables, while analytic gradients scale much better with many
* * *
[1]AEP in this paper is calculated with directionally averaged wind speeds

design variables[11]. For small wind farms and few design variables gradient-free methods are acceptable during optimization, as demonstrated in the studies mentioned. However, as the wind farm size increases and the number of design variables increases, gradient-based optimization with analytic gradients becomes necessary[5].

Differing from previous studies, in our research we used gradient-based optimization with analytic gradients for coupled optimization of many design variables. In this study, we optimize wind farms with up to 567 variables, (hub height, 3 tower diameter variables, and 3 tower thickness variables for 81 different turbines), and our model has been used to optimize over 5000 more coupled variables when we include yaw control. Our work is also unique in that we explore the impact of tower sizing along with the tower height during optimization. Taller towers need a larger diameter and shell thickness because of the associated greater forces and moments, resulting in more mass and higher costs. While having towers with different heights is beneficial in that it will reduce wake interference and produce more energy, the additional tower height of some of the turbines will increase cost. We quantify this relationship between energy production and added mass by optimizing COE, exploring when it is beneficial to have two height groups, and what the tower heights should be for each group. Finally, we examine any additional benefits from having more than two different turbine heights, and even allowing each to vary independently.

The methodology we present in this research specifically applies to onshore wind turbines. We assume that the tower height is the same as the turbine hub height in the loads and cost models. However, by appropriately considering the offshore support structure, and using offshore cost and load models, this study can be applied to offshore wind farms as well. We expect that the results we found for onshore wind farms would be similar to those of an offshore farm, namely that wind farms with mixed hub height wind turbines can result in lower COE.

**2 | METHODOLOGY**

In this section, we describe the model used to predict the COE of a wind farm. First, the wake model is discussed, which is needed to calculate the wind speed at any point in the wind farm. Next, we discuss the AEP and how it is calculated. We will then describe the structural calculations that were made for each turbine, which were used to constrain the height, diameter, and thickness of the turbines. Finally, we introduce our cost model, and how each of these components is used in optimization.

**2.1 | Wake Model**

To calculate the effective wind speed at each turbine, we used a modified version of the FLOw Redirection and Induction in Steady State (FLORIS) wake model originally presented by Gebraad et al[4]. The FLORIS wake model is derived from the Jensen model[12], but rather than use one speed to describe the wind across the wake, three separate zones are defined, each with a different expansion and decay rate. A simple weighted average of the wake overlap is used between zones to define the total effective wind speed at each turbine.

For this study, we made two modifications to the original FLORIS model. First, we used recent work that has improved FLORIS to provide a smooth response and analytic gradients[13]. These improvements enable solutions to be found for large optimization problems and help to achieve reliable answers. Second, because the original FLORIS wake model was designed to describe wakes in the horizontal plane, it was modified to calculate the effective wind speed at any point in three-dimensional (3D) space. We assume that the wake is axisymmetric, such that any cross section is circular. Additionally, we continue the assumption from the original FLORIS model that the wake center neither ascends nor descends but remains at the same height at which it originated.

A real wake may move in the vertical plane and may not maintain a perfectly circular cross section. To validate the assumptions made, we compared the model results to Simulator fOr Wind Farm Applications (SOWFA). SOWFA is a high-fidelity large eddy simulation tool that was developed at the National Renewable Energy Laboratory (NREL) for wind farm studies[14,15,16], and has been used extensively in previous research[4,17,18]. For more details on this validation, see the work by Stanley et al.[19]

**2.2 | Annual Energy Production Calculation**

The instantaneous power production of a wind farm is dependent on the wind direction, because of the wakes created behind wind turbines. For this reason, AEP is a much better indicator of a productive farm than power. Wind farm AEP takes into account the power production for all wind speeds and directions as well as the associated frequencies. The wind direction frequency

and wind speed data used in this study are from the NoordzeeWind meteorological mast located in the North Sea, measured for one year from July 2005–June 2006[20]. As seen in the left subfigure of Fig. 1, the direction frequency data is binned into 5° increments with the dominant wind direction being from the southwest. The right subfigure of Fig. 1 shows the directionally averaged wind speeds that were used to calculate AEP in this study.

[Figure]

[Figure]

**FIGURE 1** On the left is the wind direction frequency distribution from the NoordzeeWind meteorological mast. The data are divided into 72 bins. As can be seen, the predominant wind direction is from the southwest. On the right are the directionally averaged wind speeds, again for the NoordzeeWind meteorological mast.

To account for height differences affecting inflow velocity, we adjusted the wind speed data for wind shear. We used the following power law to estimate the wind speed at different heights

$$U(h) = U_{ref} \left( \frac{h}{h_{ref}} \right)^{\alpha} \tag{1}$$

where $U(h)$ is the wind speed at any height $h$, $U_{ref}$ is the reference wind speed (from the Princess Amalia wind data), the reference height $h_{ref}$ is 50 meters, and the shear coefficient $\alpha$ was varied as will be discussed later.

**2.3 | Tower Model**

Because the tower height was allowed to vary, we included a model to calculate the tower mass and perform structural analysis. The structural analysis was used to constrain the optimization to avoid failure from stress or buckling. For each tower, we calculated values and gradients of von Mises stress, shell buckling, global buckling, tower taper ratio, and the first natural frequency of the entire structure.

The tower mass was calculated from the volume of the tower, and the gradients were calculated analytically. We computed shell buckling as a function of the tower geometry and the stresses at each location, following the method outlined in Eurocode 3[21]. These calculations were made in Fortran 90, and exact gradients were obtained with the Tapenade automatic differentiation tool[22]. We performed a simple frequency calculation by approximating the tower as a cantilever beam of constant cross section with an end mass. We used the method described by Erturk et al. to calculate the natural frequency[23]. Because the turbine tower does not have a constant mass density along the length and the mass from the rotor nacelle assembly is slightly offset at the top, our frequency calculation is slightly more conservative than that predicted by a finite element model by about 10%. For this reason we scaled our frequency calculation by 10% to more closely match reality. We chose this simplified model so that we could find exact gradients, which were obtained using analytic sensitivity equations around the implicit function.

Many calculations in the structural model required the forces and moments caused by the rotor on the top of the tower. In this study, we examine two different rotor sizes. The first rotor had a 126.4 meter diameter, and we used data from the NREL 5-MW

reference turbine for the forces and moments. The second, smaller rotor has a diameter of 70 meters. For the forces and moments of this rotor, we used data from the 1.5–MW NREL WindPACT turbine, which also has a rotor diameter of 70 meters [24].

**2.4 | Cost Model**

AEP is a standard objective in wind farm optimization problems because it is easy to calculate. It is a valid measure when power production is the only dependent variable, but when the tower heights vary, AEP is no longer an appropriate measure. Changing tower height does vary AEP, but it also affects turbine capital cost. To accurately represent this relationship, we evaluated our wind farm by its COE.

COE is defined as:

$$\text{COE} = \frac{\text{FCR}[\text{TCC}(z_i, \vec{d}_i, \vec{t}_i) + \text{BOS}(z_i)] + \text{O\&M}(z_i)}{\text{AEP}(z_i)} \tag{2}$$

where the numerator represents the yearly cost of the wind farm. The FCR is the fixed charge rate, TCC is the turbine capital cost, BOS is the balance-of-station costs, and O&M is the operation and maintenance cost. TCC is a combination of the rotor, nacelle, and tower costs. The rotor and nacelle costs are taken from the turbines after which they are modeled. We approximated the tower cost as a function of tower mass at \$3.08 per kilogram. The BOS costs were calculated using Plant_CostsSE, a model created by NREL to calculate the costs of a wind farm [25]. In this model, BOS costs are slightly a function of turbine height. Operation and maintenance costs scaled with AEP, and were therefore an indirect function of $z_i$ [26].

**2.5 | Optimization**

The purpose of this study was to minimize COE of a wind farm. First, we defined the idea of a "height group." All turbines in a height group had the same tower height, diameter, and shell thickness. We parameterized the tower by specifying the diameter and shell thickness at the bottom, midpoint, and top of the tower and then linearly interpolated diameter and shell thickness at points in between. In this study, there was an equal number of turbines in each height group, or if the number of turbines in the farm was not evenly divisible by the number of groups, some groups had one additional turbine. More height groups are theoretically beneficial because they mean additional freedom in arranging the turbines to decrease wake interference. In Sect. 3.3 we explore the benefits of more than one height group, and how the number of height groups affected the optimized COE.

It may be beneficial to do a discrete optimization in which each turbine can change the height group to which it belongs, but this greatly increases the complexity of the optimization and makes it gradient-free. Discrete variables such as turbine group assignment have no intermediate values, meaning there are no gradients in their optimization. To maintain the gradient-based optimization, we assigned each turbine to one of the height groups before starting the optimization. Once assigned, a turbine could not switch to the other group.

The results of gradient-based optimization, for problems with many local minima, are sensitive to the starting location. As in most optimization problems, there is no guarantee that the solution is the true global solution. However, good results can be achieved with a multiple-start approach, where several different starting points are used for each condition, and the best solution is used. For our study we made hundreds of different random starts. While this is certainly no guarantee that we have found the true global minimum COE value of each condition, the large number of restarts lend confidence that these are good solutions with accurate trends.

Many variables affect the COE of a wind farm, with some more influential than others. In our optimization we considered the following design variables: the tower height of each group ($H_j$), the tower diameter of each group ($d_{j,k}$), and the tower shell thickness of each group ($t_{j,k}$). Index k refers to the location on the tower (k=1 is at the bottom, k=2 at the midpoint, k=3 at the top).

The tower heights were constrained to be taller than the rotor radius plus a ground clearance, which we set as 10 m. The tower diameter was constrained to be less than 6.3 m for transportation and greater than or equal to 3.87 m at the top, to allow for the connection to the nacelle. Each tower was also structurally constrained by the shell buckling and natural frequency of the tower. The shell buckling constraint was applied to each height group for both the maximum thrust conditions and the survival load, with a safety factor of 1.35 for the loads and 1.1 for buckling resistance. The first natural frequency of the tower was constrained to be greater than the frequency at which the blades rotate and less than the blade passing frequency, with a factor of safety

of 1.1. The diameter-to-thickness ratio was constrained to be greater than 120 at any point, to allow for welding during turbine assembly. The optimization can be expressed as:

$$
\begin{aligned}
\text{minimize} \quad & \text{COE} \\
\text{w.r.t.} \quad & H_j, \ d_{j,k}, \ t_{j,k} \\
& i = 1, \dots, \text{number of turbines}; \ j = 1, \dots, \text{number of height groups}; k = 1, 2, 3 \\
\text{subject to} \quad & H_j \geq r_{\text{turbine}} + 10 \text{ m} \\
& d_{j,k} \leq 6.3 \text{ m} \\
& d_{j,3)} \geq 3.87 \text{ m} \\
& \frac{3\,\Omega}{1.1} \geq f_j \geq 1.1\,\Omega \\
& \text{shell buckling margins: max thrust} \leq 1 \\
& \text{shell buckling margins: survival load} \leq 1 \\
& \frac{d_{j,k}}{t_{j,k}} \geq 120
\end{aligned}
\tag{3}
$$

Note that $i$ is the index defining the wind turbine, $j$ is the index describing the height group, and k describes the location on the tower.

The gradients for this optimization were all analytic. We calculated the partial derivatives of each small section of the model and included each part in a framework called OpenMDAO[27], which calculates the gradients of the entire system. Each optimization was performed using SNOPT (Sparse Nonlinear OPTimizer), a gradient-based optimization algorithm that works well with problems that have high dimensionality[28].

**3 | RESULTS**

As shown by Vasel-Be-Hagh and Archer, different hub heights are more advantageous in certain conditions[10]. In our study, we explored how turbine spacing, wind shear exponent, and rotor diameter affect the optimized COE of wind farms with different hub heights, compared to wind farms with all the same height.

To compare the results, we ran several different optimization cases: a baseline grid wind with no height or layout optimization, a grid with one height group, and a grid with two height groups. These optimizations, their design variables, and how they are described throughout the rest of the paper are shown in Table 1.

When there was no height optimization, the hub heights were set at 90 meters, which is the hub height of the NREL 5-MW reference turbine. For each of the situations listed we optimized two farms, one farm with 25 5-MW wind turbines with 126.4-meter rotor diameters (the large rotor wind farm), and one farm with 81 70-meter rotor diameter turbines (the small rotor wind farm). These rotor diameters were chosen because each is associated with an NREL reference turbine for which we already have rotor load data. The turbines in the small rotor wind farm each had a capacity of 1.543-MW, making the total capacity 125 MW in both the large rotor and small rotor wind farm. The rated wind speed was 11.4 m/s for both the large and small rotors. We idealized the wind turbines with a cut-in wind speed of zero, meaning the turbines produced power for any non-zero wind speed. Because we used directionally averaged wind speeds, the cut-out speed was not defined for either rotor, as extreme high wind speeds were never used.

**3.1 | Varied Wind Shear**

We started by optimizing the turbine design of a grid wind farm through varied wind shear. We expected that two different hub heights would be more beneficial in cases with lower wind shear, because the wind speed at lower heights is almost the same as it is higher up. This means that a lower tower will not experience a significant decrease in wind speed but will be able to reduce wake interference. We varied the wind shear exponent from 0.08 to 0.3, which are realistic extremes for average wind shear (although more extreme values do occur). While varying the wind shear exponent, we explored 2 different square wind farm sizes: the big farm was 2,800 meters by 2,800 meters, and the small farm was 1,680 meters by 1,680 meters. These wind farm

sizes and their grid spacing are summarized in Table 2. For these optimizations, we alternated the placement of height group to make a checkerboard pattern. This pattern seemed logical and produces good results.

Figures 2 and 3 show the results of optimizing the small rotor wind farms while varying the wind shear exponent. In each figure, the subfigure in the top left shows the optimized COE as a function of the wind shear exponent. The top right subfigure shows the optimized heights of each of the two height groups, corresponding to the optimization where there are two different height groups. The bottom left subfigure shows the wake loss for each of the optimized wind farms. This wake loss is defined as the percent AEP loss in the wind farm compared to ideal conditions where each turbine in the farm is always exposed to the free stream wind. The bottom right subfigure shows both the ideal AEP and true AEP values for the optimized wind farms. Ideal AEP is the AEP of the wind farm if each turbine were always exposed to free stream wind conditions. The true AEP is calculated in the wind farm, with wake interactions between turbines.

[Figure]

**FIGURE 2** Optimization results on a big wind farm, small rotor wind farm while varying the wind shear exponent. The top left figure shows the optimized COE as a function of wind shear exponent. The top right figure shows the optimized hub heights of the optimization runs with two height groups. The bottom left figure shows the percent wake loss for the optimized wind farms. The bottom right figure shows the ideal and true AEP for the optimized wind farms.

As the shear exponent increases, the benefit of two height groups over one decreases. The COE plot in Fig. 2 shows the results for the big wind farm, where the turbines are spaced the farthest apart. There is a very small benefit to having two height groups for the low shear exponents with the largest COE decrease of 2.2% compared to the optimization with one uniform height group.

[Figure]

**FIGURE 3** Optimization results on a small wind farm, small rotor wind farm while varying the wind shear exponent. The top left figure shows the optimized COE as a function of wind shear exponent. The top right figure shows the optimized hub heights of the optimization runs with two height groups. The bottom left figure shows the percent wake loss for the optimized wind farms. The bottom right figure shows the ideal and true AEP for the optimized wind farms.

This benefit decreases with increasing wind shear, and around a shear exponent of 0.22 any benefit is negligible. The top left subfigure of Fig. 3 shows the optimized COE results for the small wind farm. Here, the turbines are very tightly spaced, and there is a significant benefit to different height groups for all wind shear values. The greatest benefit is an 11.2% COE decrease compared to one height group, for a wind shear of 0.08, and 7.1% for the highest wind shear of 0.3. As we expected, two different height groups are more beneficial when there is a low wind shear exponent, which means that the wind speed does not change very quickly with height. These results also indicate that, for a grid layout, two different height groups are much more beneficial in wind farms where the turbines are close together. When the turbines are farther apart, the wakes are much weaker between turbines, meaning that there is not as much benefit from avoiding a wake. When the turbines are very close together the wakes create a very large decrease in wind speed, making it more beneficial to avoid wake interference.

The top right subfigures in Figs. 2 and 3 show the optimized heights of the two different height groups. For the low shear exponent values, there is a large separation between the two heights. This difference decreases as shear increases. In Fig. 2, the highest shear exponent value has no difference between the two height groups, indicating that for this rotor diameter, turbine spacing, and wind shear, there is no benefit to having different height groups. The small wind farm results has a greater difference between the heights of each turbine group for all shear exponents. For the small wind farm, the shorter height group remains at

the minimum (45 m) up through a shear exponent of 0.15. The big wind farm has the shorter height group at the minimum for only a few of the lowest shear exponents.

In Figs. 2 and 3, the bottom left subfigure shows the percent AEP loss due to turbine wakes in each optimized wind farm, and the bottom right subfigure shows both the ideal and true AEP values for each of the optimized wind farms. In Fig. 2, for the big wind farm, the wind farm with two groups has lower wake loss that the baseline farm and wind farm with 1 group, up to about a shear exponent of 0.22. After this, there is little to no reduction in wake loss, corresponding to the small separation in optimized hub heights and negligible reduction in optimized COE. Also in Fig. 2, we see that the true wind farm AEP of the wind farms with optimized turbine heights is slightly lower than the baseline case, meaning that the lower optimized COE is a result of lower turbine costs instead of higher AEP. In Fig. 3, for the small wind farm, the wind farms with two different height groups have more than 10% less wake loss than the baseline wind farm and wind farm with one height group. This is from the large separation in the optimized hub heights. In the AEP plot, the ideal AEP of the wind farm with two different heights is lower than the ideal AEP with 1 height group and the baseline case. If these turbines were all exposed to the free stream wind conditions, they would perform poorly compared to the other turbine designs. However, in the wind farm, these turbines perform better than the other turbine heights. The lower COE from the wind farms with two different height groups are caused by higher AEP, in addition to potentially lower turbine costs.

Figures 4 and 5 show the results of optimizing a grid, large rotor wind farm while varying the wind shear exponent. Again as in Figs. 2 and 3, the subfigures in the top left show the optimized COE as a function of the wind shear exponent. The top right subfigures show the optimized heights of each of the two height groups, corresponding to the optimization where there are two different height groups. The bottom left subfigures show the wake loss for each of the optimized wind farms. The bottom right subfigures show the ideal AEP and true AEP values for the optimized wind farms.

The trends of the large rotor farm optimizations differ from the small rotor farm. The big farm shows negligible, if any, benefit between the COE for one and two height groups. For the big wind farm, the largest benefit for two height groups is a mere 0.04% at the shear value of 0.12. Even compared to the baseline case, any height optimization results in only a very small COE decrease. The optimized hub heights plot shows that even when two different hub heights are allowed, for almost all shear exponents, it is optimal for the turbines to be the same height. For the big wind farm, the most interesting plot is the AEP, where we see that for the higher shear exponents (above 0.15), optimizing the turbine heights results in a higher AEP. Thus, for a similar COE, more energy can be produced.

For the small wind farm results in Fig. 5, there is a small but noticeable benefit between one and two height groups in the shear exponent range 0.12-0.18. The largest benefit is 1.2% at the 0.14 shear exponent. For this same range of shear exponents, there is a noticeable difference in the optimized hub heights of each group, as well as a noticeable decrease in the percent wake loss of the wind farms with two height groups. Again, as with the big wind farm, the wind farms with optimized hub heights result in a higher AEP for shear exponents above 0.15.

The benefits from two different height groups are about ten times greater for the small rotor wind farm than the large rotor farm. The major reason for this difference is the relative size of the rotor compared to the height of the turbine. For the small wind farms with a shear exponent of 0.15, the height groups are greatly separated for both the big and small wind farms (see the top right subfigures of Figs. 3 and 5). The small rotor wind farm has a height difference of just over 50 m (Fig. 3), and the big rotor wind farm has a height difference of about 33 m. Figure 6 shows what these height differences actually look like for the different rotor diameters. For the small rotor case, the 50 m difference in hub heights results in very little overlap between the rotors of the different height groups. Even when directly in line with the wind direction, the wakes from one height group will not greatly affect the turbines in the other group. Compare this to the large rotor farm. Even at maximum separation, there is a large overlap between the rotors of each height group. The wakes of each group will always greatly impact the power production of the other.

**3.2  |  **Varied Turbine Spacing**

Next, we explored how different height groups benefit farms with different turbine spacing. We investigated this with a metric called turbine density. Turbine density is a measure of the average turbine spacing in the wind farm and is defined as the area of all of the rotor disks relative to the total farm area:

$$\text{Turbine Density} = \frac{\pi R^2 N}{A} \tag{4}$$

[Figure]

**FIGURE 4** Optimization results on a big wind farm, big rotor wind farm while varying the wind shear exponent. The top left figure shows the optimized COE as a function of wind shear exponent. The top right figure shows the optimized hub heights of the optimization runs with two height groups. The bottom left figure shows the percent wake loss for the optimized wind farms. The bottom right figure shows the ideal and true AEP for the optimized wind farms.

where $R$ is the rotor radius, $N$ is the number of turbines, and $A$ is the area of the wind farm. We varied the turbine density by changing the total area of the wind farm, while keeping the number of turbines and their rotor diameter the same. For this portion of the study, we ran density sweeps with two shear exponent values: 0.08 and 0.25.

Figures 7 and 8 show the results of our optimization runs while varying the turbine density for a grid wind farm with a small rotor and a shear exponent of 0.08 and 0.25, respectively. Each of the subfigures shows the same information as Figs. 2–5 with the varied wind shear. The top left shows COE, the top right shows optimized hub heights, the bottom left shows wake loss, and the bottom right shows ideal and true AEP.

In Figs. 7 and 8, it is apparent that at high turbine densities, two height groups have a much lower COE than one height group. As the turbine density decreases, this benefit also decreases. Also, the lower shear exponent of 0.08 in Fig. 7 has greater COE reduction with two height groups than a shear exponent of 0.25 in Fig. 8. In Fig. 7, at a shear exponent of 0.08 and at the highest turbine density, there is a 23.3% decrease in COE from one height group to two. In Fig. 8, at a shear exponent of 0.25 and the highest turbine density, the COE decrease is 20.6%. Also notice the COE of one height group is identical to the baseline grid at the shear exponent of 0.25. This occurs because the optimized height of the one height group approaches the baseline height of 90 meters.

[Figure]

**FIGURE 5** Optimization results on a small wind farm, big rotor wind farm while varying the wind shear exponent. The top left figure shows the optimized COE as a function of wind shear exponent. The top right figure shows the optimized hub heights of the optimization runs with two height groups. The bottom left figure shows the percent wake loss for the optimized wind farms. The bottom right figure shows the ideal and true AEP for the optimized wind farms.

The top right subfigures of Figs. 8 and 9 show the optimized hub heights of the two height groups. For both shear exponents and for almost all turbine density values, there is a large difference between the two height groups (over 50 meters). Only below a turbine density of 0.1 do the hub heights get significantly closer together, for both shear values.

The bottom subfigures of Figs. 8 and 9 show the wake loss in each optimized wind farm and the ideal and true AEP values for each optimized wind farm. For each shear exponent, these figures are very similar. With two height groups, there is much lower wake loss than when there is only one height group. This is true across all the turbine densities except for the very lowest, when the turbines are very far apart (close to 10 rotor diameters). Also, as in Fig. 3, across almost all turbine densities the ideal AEP is lower with two height groups, but the true AEP in the wind farm is higher. The only exception is in Fig. 8 for turbine densities below 0.05. Again, this means that for the small rotor diameter, wind farms with different hub heights can produce a higher AEP, and have a lower COE.

Wind farms with high turbine density have turbines that are very close together, meaning that there is significant wake interference. For these farms, there is a benefit to having different hub heights, to avoid wakes between turbines. At low turbine density the turbines are farther apart, meaning that the wakes are much weaker and the wind speed is higher. In these situations, having two height groups is not as beneficial.

**Optimized Wind Turbine Heights**

[Figure]

[Figure]

**FIGURE 6** The two optimized height groups for the small wind farm at a shear exponent of 0.15. On the left is one turbine from each height group in the small rotor wind farm; on the right is one turbine from each height group of the big rotor wind farm.

Figures 9 and 10 show the results of our optimization runs while varying both the turbine density for a grid wind farm with a large rotor and shear exponents of 0.08 and 0.25, respectively. Again, the top left of each figure shows COE, the top right shows optimized hub heights, the bottom left shows wake loss, and the bottom right shows ideal and true AEP.

These results behaved differently than originally expected. While the benefit of two height groups over one does increase with increasing turbine density for the large rotor wind farm, it is not fully realized at the lowest shear exponent, 0.08. For both shear exponents, 0.08 and 0.25, the benefit of having two height groups over one at the highest turbine density is 0.7% and 2.5%, respectively. At a shear exponent of 0.08 in Fig. 9, there is a small benefit to having two height groups only at the very highest turbine densities (0.225–0.25). For a shear value of 0.25 in Fig. 10, there is a significant benefit to two height groups, but only down to a turbine density of around 0.175. The reason for this behavior is explained in the top right subfigures of Figs. 9 and 10, which show the optimized turbine heights. In Fig. 9, the tower heights of both groups are at the height minimum (73.2 meters) for the majority of turbine densities. At this shear exponent of 0.08, the higher wind speed and power production gained from a higher tower is not worth the extra cost. In fact, it would result in a lower COE to have a shorter tower, but this value is constrained at 73.2 meters to allow sufficient clearance between the ground and the blades. Now contrast this behavior with Fig. 10. Through a turbine density of 0.125 there is no separation between the height groups. However, below this turbine density the optimized height does not go to the minimum but remains much higher, over 100 meters. At this shear exponent and turbine densities, it is not worth the decrease in wind speed to have one shorter height group.

When turbine heights are optimized coupled with turbine layout or yaw there is a clear benefit from the additional design variables; however, the trends for the benefit of two height groups remain the same. Every point in the COE portions of Figs. 2–5 and 6–10 shifts down approximately equally with the other points in the same figure, keeping the benefit of two height groups over one the same for each case. When layout and yaw control are also included as design variables, two height groups are still more beneficial in wind farms with low wind shear, high turbine densities, and small rotor diameters.

**3.3 | Effect of the Number of Height Groups**

Up to this point, the highest number of height groups in a farm has been two. However, if there is a large decrease in COE from one height group to two, it is logical to think there could be additional benefits to adding more height groups, or even letting each turbine vary its height individually. This would add additional complexity to the design and manufacture of a wind farm, but large COE reduction could be worth the additional complexity.

To study the effects of additional height groups, we optimized the small wind farm with a grid layout, and the wind shear exponent at 0.15. We then varied the number of height groups in the optimization from 1 up to the number of turbines in the farm. Figure 11 shows the results for optimizing the wind farms with different numbers of height groups. On the left are the results for the small rotor wind farm, while on the right are the results for the large rotor farm.

[Figure]

**FIGURE 7** Optimization results of a small rotor wind farm while varying the turbine density with a wind shear exponent of 0.08. The top left figure shows the optimized COE as a function of wind shear exponent. The top right figure shows the optimized hub heights of the optimization runs with two height groups. The bottom left figure shows the percent wake loss for the optimized wind farms. The bottom right figure shows the ideal and true AEP for the optimized wind farms.

There is a large COE decrease when optimizing with two height groups as compared to just one; however, adding more height groups after that is not tremendously beneficial. In Fig. 11 on the left, for the small rotor farm, there is a 13.5% COE decrease from one height group to two and only a 1.0% decrease from 2 height groups to 81, where each turbine can vary independently. On the right, for the large rotor farm, there is a 1.6% decrease in COE from one height group to two and a 0.07% decrease from 2 height groups to 25. From one height group to two, the large COE reduction appears to be worth the small added complexity in wind farm design and manufacturing. However, adding more height groups or letting each turbine be optimized individually does not seem to be worth it.

**4 | CONCLUSIONS**

This research presented a method to optimize a wind farm with different hub heights using exact analytic gradients and structural constraints. We discuss how to account for wake interference between wind turbines of different hub heights, wind shear, structural calculations to constrain the tower during optimization, and the cost of the wind farm. This paper is unique compared to other research in this area in that we use structural constraints, and sized the entire tower. We explored how the increased

[Figure]

**FIGURE 8** Optimization results of a small rotor wind farm while varying the turbine density with a wind shear exponent of 0.25. The top left figure shows the optimized COE as a function of wind shear exponent. The top right figure shows the optimized hub heights of the optimization runs with two height groups. The bottom left figure shows the percent wake loss for the optimized wind farms. The bottom right figure shows the ideal and true AEP for the optimized wind farms.

AEP from different hub heights interacts with the additional costs from taller towers by optimizing COE. It is also unique in that we used a gradient-based approach, which allowed us to optimize much larger farms with more design variables than are feasible with gradient-free optimization. The most variables that we optimized in this research were hub height, tower diameter, and tower shell thickness for an 81-turbine wind farm with each turbine varying it individually, a total of 567 design variables.

From our optimization results, we conclude that wind farms with two different hub heights are beneficial in wind farms with small rotor diameters relative to the maximum height of the tower, low wind shear, and high turbine density. For a grid wind farm with 81 turbines, 70 meter rotor diameters, a wind shear exponent of 0.08, and 4 diameter spacing between turbines, there is a 4.9% decrease in cost of energy when using two height groups instead of one. For the same farm with 3 diameter spacing between turbines, there is an 11.2% decrease in COE when two height groups are included. A small rotor diameter allows greater separation between the rotors of different height groups, and minimizes wake interactions between height groups. Low wind shear means similar wind speeds close to the ground as well as higher up, so a shorter height group does not experience significantly lower wind speeds. Farms with high turbine density have tightly packed turbines, meaning there is a large wind velocity deficit in the wakes. In these cases, it is more effective to have different turbine heights to reduce this wake interference.

[Figure]

**FIGURE 9** Optimization results of a big rotor wind farm while varying the turbine density with a wind shear exponent of 0.08. The top left figure shows the optimized COE as a function of wind shear exponent. The top right figure shows the optimized hub heights of the optimization runs with two height groups. The bottom left figure shows the percent wake loss for the optimized wind farms. The bottom right figure shows the ideal and true AEP for the optimized wind farms.

There is a large benefit when comparing two height groups to one; however, using more height groups does not provide a significant reduction in COE. The benefit from two height groups to letting every turbine change individually is negligible compared to going from one height group to two. In our opinion, the small benefit is not worth the added complexity.

We suggest and will pursue several options to continue this research. First is to include other turbine differences beyond hub height. This could include different rotor diameters, turbine ratings, turbine types, blade numbers, or several other differences. We expect including other turbine differences in the same wind farm to further decrease wind farm COE. Another continuation of this study will be to further explore the height group turbine assignments and distributions. In this study we assign roughly half of the turbines in the farm to be in each height group. There are likely better ways to assign turbines to a certain height group, and it may be optimal to have more turbines in one height group than the other. Also, because we used basic cylindrical steel tubes, the structural constraints prevented the towers in the large rotor farm to be tall enough for significant height separation between groups. We suggest exploring larger tower diameters and more advanced tower designs to allow the large rotor towers to be much taller, such that different tower heights may be more beneficial. Finally, in future work it may be beneficial to tune the FLORIS wake model parameters for onshore conditions. In this work, we used parameters that had been tuned for the Princess Amalia wind farm, an offshore farm. Future work should include FLORIS parameters that have been tuned specifically for the wind farms they are modeling.

[Figure]

**FIGURE 10** Optimization results of a big rotor wind farm while varying the turbine density with a wind shear exponent of 0.25. The top left figure shows the optimized COE as a function of wind shear exponent. The top right figure shows the optimized hub heights of the optimization runs with two height groups. The bottom left figure shows the percent wake loss for the optimized wind farms. The bottom right figure shows the ideal and true AEP for the optimized wind farms.

**ACKNOWLEDGMENTS**

The BYU authors developed this journal article based on funding from the Alliance for Sustainable Energy, LLC, Managing and Operating Contractor for the National Renewable Energy Laboratory for the U.S. Department of Energy.

This work was authored [in part] by the National Renewable Energy Laboratory, operated by Alliance for Sustainable Energy, LLC, for the U.S. Department of Energy (DOE) under Contract No. DE-AC36-08GO28308. Funding provided by the U.S. Department of Energy Office of Energy Efficiency and Renewable Energy Wind Energy Technologies Office. The views expressed in the article do not necessarily represent the views of the DOE or the U.S. Government.

The U.S. Government retains and the publisher, by accepting the article for publication, acknowledges that the U.S. Government retains a nonexclusive, paid-up, irrevocable, worldwide license to publish or reproduce the published form of this work, or allow others to do so, for U.S. Government purposes.

[Figure]

**FIGURE 11** The effect of optimized COE as a function of the number of height groups. On the left are results for the small rotor wind farm, on the right are results for the large rotor farm. These are for the small wind farm, and a shear exponent of 0.15.

**References**

1. Kusiak A, Song Z. Design of Wind Farm Layout for Maximum Wind Energy Capture. *Renewable Energy* 2010; 35(3): 685–694. https://doi.org/10.1016/j.renene.2009.08.019.

2. Şişbot S, Turgut Ö, Tunç M, Çamdalı Ü. Optimal Positioning of Wind Turbines on Gökçeada Using Multi-Objective Genetic Algorithm. *Wind Energy* 2010; 13(4): 297–306. doi:10.1002/we.339.

3. Wagner M, Veeramachaneni K, Neumann F, O'Reilly UM. Optimizing the layout of 1000 wind turbines. *European wind energy association annual event* 2011: 205–209.

4. Gebraad PMO, Teeuwisse FW, Wingerden vJW, et al. Wind Plant Power Optimization Through Yaw Control Using a Parametric Model for Wake Effects – a CFD Simulation Study. *Wind Energy* 2016; 19(1): 95–114. doi:10.1002/we.1822.

5. Fleming P, Ning A, Gebraad P, Dykes K. Wind Plant System Engineering through Optimization of Layout and Yaw Control. *Wind Energy* 2016; 19(2): 329–344. doi:10.1002/we.1836.

6. Graf P, Dykes K, Scott G, et al. Wind farm turbine type and placement optimization. *Journal of Physics: Conference Series* 2016; 753(062004).

7. Chen Y, Li H, Jin K, Song Q. Wind Farm Layout Optimization Using Genetic Algorithm with Different Hub Height Wind Turbines. *Energy Conversion and Management* 2013; 70: 56–65. doi:10.1016/j.enconman.2013.02.007.

8. Hazra J, Mitra S, Mathew S, Zaini F. 3D Layout Optimization for Large Wind Farms. In: ISGT. ; 2015: 1-5. http://dx.doi.org/10.1109/ISGT.2015.7131819.

9. Chen K, Song M, Zhang X, Wang S. Wind turbine layout optimization with multiple hub height wind turbines using greedy algorithm. *Renewable Energy* 2016; 96: 676–686. https://doi.org/10.1016/j.renene.2016.05.018.

10. Vasel-Be-Hagh A, Archer CL. Wind farm hub height optimization. *Applied Energy* 2017; 195: 905–921. https://doi.org/10.1016/j.apenergy.2017.03.089.

11. Ning A, Petch D. Integrated design of downwind land-based wind turbines using analytic gradients. *Wind Energy* 2016; 19(12): 2137–2152. doi:10.1002/we.1972.

12. Jensen NO. *A note on wind generator interaction* . 1983.

13. Thomas J, Gebraad P, Ning A. Improving the FLORIS Wind Plant Model for Compatibility with Gradient-based Optimization. *Wind Engineering* 2017; doi:10.1002/we.1822.

14. Churchfield M, Lee S. NWTC design codes (SOWFA), 2012. .

15. Jonkman J. NWTC design codes (FAST). *NWTC Design Codes (FAST), NREL, Boulder, CO* 2010.

16. Churchfield MJ, Lee S, Michalakes J, Moriarty PJ. A Numerical Study of the Effects of Atmospheric and Wake Turbulence on Wind Turbine Dynamics. *Journal of turbulence* 2012(13): N14. http://dx.doi.org/10.1080/14685248.2012.668191.

17. Fleming PA, Gebraad PM, Lee S, et al. Evaluating Techniques for Redirecting Turbine Wakes Using SOWFA. *Renewable Energy* 2014; 70: 211–218. https://doi.org/10.1016/j.renene.2014.02.015.

18. Fleming P, Gebraad PM, Lee S, et al. Simulation Comparison of Wake Mitigation Control Strategies for a Two-Turbine Case. *Wind Energy* 2015; 18(12): 2135–2143. doi:10.1002/we.1810.

19. Stanley AP, Thomas J, Ning A, Dykes K, Fleming PA. Gradient Based Optimization of Wind Farms with Multiple Turbine Hub Heights. In: 35th Wind Energy Symposium. ; 2017: 16–19. doi:10.2514/6.2017-1619.

20. NoordzeeWind B. Data of the NoordzeeWind monitoring and evaluation programme (NSW-MEP). 2013.

21. *Eurocode 3: design of steel structures : part 1-5 : plated structural elements*. London: British Standards Institution . 2010. Incorporating corrigendum April 2009.

22. Hascoet L, Pascual V. The Tapenade Automatic Differentiation Tool: Principles, Model, and Specification. *ACM Transactions on Mathematical Software (TOMS)* 2013; 39(3): 20. doi:10.1145/2450153.2450158.

23. Erturk A, Inman DJ. Appendix C: Modal Analysis of a Uniform Cantilever With a Tip Mass. *Piezoelectric Energy Harvesting* 2011: 353–366. doi:10.1002/9781119991151.app3.

24. Malcolm D, Hansen A. WindPACT turbine rotor design study. *National Renewable Energy Laboratory, Golden, CO* 2002; 5. doi:10.2172/15000964.

25. Dykes K, Scott G. Plant_CostsSE Documentation version 1.0. *NREL TP forthcoming* 2014.

26. Moné C, Smith A, Maples B, Hand M. Cost of Wind Energy Review. tech. rep., NREL/TP-5000-63267. Golden, Colorado: National Renewable Energy Laboratory; 2013.

27. Gray J, Moore K, Naylor B. OpenMDAO: An open source framework for multidisciplinary analysis and optimization. 2010.

28. Gill PE, Murray W, Saunders MA. SNOPT: An SQP Algorithm for Large-Scale Constrained Optimization. 2002; 12(4): 979–1006. Updated version appeared in SIAM Review 47(1), 2005, pages 99–131. https://doi.org/10.1137/S0036144504446096.

**TABLE 1** A list of optimizations run in this study. On the left are the names by which they are described throughout the figures and rest of the paper; on the right are the design variables included in each run.

| Optimization | Design Variables |
| --- | --- |
| baseline | $\vec{d}_1, \vec{t}_1$ |
| 1 group | $\vec{d}_1, \vec{t}_1, H_1$ |
| 2 groups | $\vec{d}_{1,2}, \vec{t}_{1,2}, H_{1,2}$ |

**TABLE 2** The sizes of the big and small wind farms used in optimization. Sizes are shown absolutely, as well as the grid spacing in rotor diameters.

| Wind Farm | Size (m) | Grid Spacing (Small Rotor Diameters) | Grid Spacing (Big Rotor Diameters) |
|-----------|----------|:---:|:---:|
| big | 2800 by 2800 | 5 | ≈ 5.5 |
| small | 1680 by 1680 | 3 | ≈ 3.3 |

---

## Author Comment (AC3) · 30 Oct 2018

**Response to Reviewer 2**

Andrew PJ Stanley and Andrew Ning

October 30, 2018

Thank you for your comments and questions! We will respond to each of your comments/questions below, individually. In addition to the comments below there were a few small wording change suggestions, all of which were made in the updated document.

Question/Comments are in black.

The corresponding responses are immediately below in blue.

**Page 1**

Rotor diameter and rated power often closely correlated! ... and can in general not be considered as independent variables. Please motivate why both are included in your design space.

There is certainly some correlation between rotor diameter and rated power, however they are not completely coupled. Take for example the NREL 5-MW reference turbine

(rotor diameter 126.4 meters) and the IEA37 3.35-MW onshore reference turbine (rotor diameter 130 meters). Both have very different power ratings but almost identical rotor diameters. For this reason we considered them as independent variables.

What is meant here - please explain. Is blade chord and twist distribution included in the design space?

This was clarified in the text a paragraph before equation 6.

Yes - maybe not surprising as sequential optimization can be considered as a subset of integrated optimization?

True, this is not the most surprising conclusion.

and increasing the loading

This change was included in the updated document.

Exact analytical gradients ... but from a surrogate model. Analytical gradients of a surrogate model may differ from true gradients!

Exact analytic gradients refers to the whole model, including the wake model, power calculations, tower structural model, wind speed distribution, cost model, and the rotor/nacelle surrogate model.

... but is more easily trapped in local extrema than some gradient-free approaches!

This is mentioned in line 24: "...and while gradient-based optimization methods converge to local minima..."

[Figure]

can be compensated to some degree by a two-step procedure where coarse griding of a gradient-free approach is combined with a gradient-based optimizer

Excellent point! This optimization approach could be considered in future research.

**Page 2**

random search?

Yes. Random starting points.

wind turbine de-rating and active wind turbine yaw control

This change was made in the updated document.

A mix of gradient free and gradient based optimization approaches might also be advantageous as e.g. in "TOPFARM: Multi-fidelity optimization of wind farms", Pierre-Elouan Réthoré et al., WE, 2013.

This change was included in the updated document.

well - in principle simple. However, wake interaction as a concept is not unambiguous. Does it cover only power losses ... or does it also include increased (fatigue) loading, which will increase cost of operation and maybe in the end compromise the life-time of the wind farm.

Yes! The principle is simple but becomes challenging in practice.

Thus, it makes sense to base the layout optimization on distributions averaged over the life-time of the wind farm ... and to consider sensitivity of the optimized results based on distribution uncertainty!

Correct.

I consider these two parameters highly correlated. Please explain why you have included both in the design space.

Responded to above in a comment from page 1.

... but possible increased cost associated with use of a multitude of tower heights is also a part of the COE equation!

The turbine costs were considered in the COE equation. We included cost of extra tower mass and the cost of the rotor diameter. As for additional costs that may come from having "custom" tower heights, we talked to a company that said there would be no additional cost just to manufacture different tower heights (beyond the extra material costs, obviously). Even if it did however, in this study we assumed these extra costs would be negligible as we only consider two different turbine designs, rather than letting each turbine vary individually.

**Page 3**

... but the wind shear is not only site dependent but can, for a particular site, depend significantly on atmospheric stability, and thereby vary over time (e.g. follow a diurnal cycle)

The updated document was clarified to say that the wind shear was assumed to be constant.

... but again to be meaningful potential additional costs related to two tower heights must be included in the 'equation'

These additional costs were included in the model.

Not demonstrated if only positive effects are included (power increase), whereas possible negative effects (increased financial costs) are not accounted for

These additional costs were included in the model.

This is a 'must' to make the study meaningful

Correct.

These refer to a surrogate model ... and does not necessarily 'collapse' with true gradients of the physical system. I tend to consider gradients of a surrogate less 'robust' than the surrogate itself.

Responded to above in a comment from page 1.

Does this model describe the 3D flow field in the wind farm?

Yes! This was clarified in the updated document.

deficit, L,

This change was made in the updated document.
Why not logarithmic law , which is physically funded for neutral atmospheric conditions? The power law is a purely empirical model

We decided to use the power law because it is easy to visualize the effect that changing the wind shear exponent has on wind speeds at different heights. However, this is not fundamental to the study, and using a logarithmic for future work is a good suggestion.

**Page 4**

Is this a top-hat type of profile? ... and what in case of partial wake situations? Further, talking about rotor effective wind speed it seems as if a linear shear over the rotor has been assumed here ... contradicting the shear defined in eq. 2

The wake profile is discussed in Section 2.1. Although we briefly discuss some of the changes that we made to the original FLORIS formulation, we do not re-present all of the details of the wake model as they are described in the citations. The partial wake situations were clarified to mention the area weighted average. The shear equation is used to define the free stream velocity at hub height, which is assumed to be constant across the rotor.

Could you please motivate this choice?

We motivated this choice to be from a range of realistic shape factors that were fit to real wind distributions.

Could you comment on the universality of this result - do you e.g. expect this result to be wind farm topology independent or not; ambient turbulence dependent (i.e. wake expansion) dependent or not, ...

Great question! The number of samples required for convergence is a function of the power with respect to wind direction for the wind farm. If this curve is smooth, the number of samples required to accurately compute expected power will be small. If it is very noisy, you will need many samples to accurately compute the expected power.

The power w.r.t. wind direction curve is determined by the wind farm layout, the wake model, and the wind distribution. The samples used in this study were determined from the baseline layout, which has many highs and lows in the power curve. We expect this to become more well behaved as an optimal layout is approached, therefore concluded that this number of samples was sufficient.

In short, no this result is not universal, but unique to our specific wind farms and wind roses.

**Page 5**

It is not straight forward to see how this is ensured - please briefly explain

This was reworded for clarity in the updated document.

Which type of surrogate model is used here ... and why? Please motivate

Mentioned later in the paragraph, 5th order bivariate spline.

**Page 6**

OK -this is the surrogate!

Correct!

I suppose you checked the accuracy of the surrogate - did you also check the accuracy of the surrogate gradients?

The surrogate is a polynomial spline fit, thus the gradients are simple and can be calculated by hand.

**Page 7**

Could also be economic performance of the wind farm taking into accounts cost of loading

Correct, we use cost of energy in this study.

... but in general power is the only aspect - cost of loads are also important

Correct. Especially in this study it is important to consider costs in addition to AEP.

In addition to financial costs, operational costs - linked to turbine loadings in the wind farm - is also of relevance. See e.g. "TOPFARM: Multi-fidelity optimization of wind farms", Pierre-Elouan Réthoré et al., WE, 2013

Operational costs are included in the cost model.

By computing all possible weightnings you should approach the binary optimization approach!

Correct! This would be an interesting continuation of this research.

**Page 8**

Again, couldn't rotor diameter and rated power be collapsed to one set of design variables?

Responded to above in a comment from page 1.

Please explain

We clarified here that implicit design variables change in the optimization process (through the rotor diameter and rated power surrogate), but are not explicitly manipulated.

**Page 9**

How is this statement justified ... given the fact the analytic gradients refer to a surrogate model?

Responded to above in a comment from page 1.

None of these is the IEC-code recommended. Why not?

We want to span the realistic range of shear exponents in a wind farm, skewing slightly towards the lower end typical over open water or flat ground. We chose 3 shear exponents that were equally spaced (0.075, 0.175, 0.275). Obviously we could have shifted all of these slightly up to consider (0.1, 0.2, and 0.3), which would include the IEC recommended 0.2, but we wanted to make sure to include the lowest value of 0.075.

I take this as a simple scaling of the topology ... but the optimal configuration might be with a completely different layout structure ... could you comment on this? You probably mean scaling of the base line wind turbine locations??

Yes the wording on this was confusing. It has been reworded for clarity.

**Page 11**

Would have been interesting with a sanity check based on the circular wind farm with a uniform direction pdf - in this case I would expect a segment of the wind turbines to be located equidistantly on the circular boundary of the wind farm

We assume here you mean to check if the layout optimization performed as expected? Of course the wake model and optimization was tested on simple cases. However, we elected to only include the most interesting wind farm layouts and wind roses in the paper.

... or by combining a generic optimization algorithm with a gradient based

This optimization approach could be considered in future research.

**Page 12**

... to the extend cable costs are not included, which is usually important for layout optimization. Without cable cost included (and no area constraints) an optimization will lead to 'clusters' of solitary wind turbines ... which is obviously not economical feasible in reality.

Excellent point. We added a paragraph at the end of Conclusions that addresses this and recommends this course of action in future research.

**Page 14**

None of these matches the IEC recommended value(s)

This was responded to above in a comment from page 9.

**Page 15**

I would expect the combined optimization to result in different turbine design depending on whether the turbine is positioned on the edge of the wind farm or deep inside the wind farm. Have you performed this 'sanity check'?

Yes, the turbine design is determined by the wind it is exposed to.

Except for the spacing constrain, I would a priory expect that low wind speeds call for large rotor diameters (i.e. low wind rotors) - could you comment on this? ... is it maybe load dictated in case of waked wind conditions??

This is surprising. Considering a solitary turbine, the consequence is that large turbines only pays off at high wind sites. Are you sure that your wind turbine cost model is realistic?

We'll address these two comments together. Remember that our objective function is COE, so simultaneously seeking to minimize cost and maximize AEP. This does **NOT** mean that the optimal designs we present are necessarily optimal for another objective, say AEP or total profit. For some objectives it would certainly be more desirable to have a large rotor diameter at low wind speeds to capture as much energy as possible or make more money, however these were not our objectives.

**Page 16**

However, if grid costs was included in the model, this conclusion might very well be completely different!

This was responded to above in a comment from page 12.

Applying different control settings of each individual wind turbine might be a cheaper way of achieving different rotor performances throughout the wind farm?

Optimizing turbine design does not replace optimal control of a wind farm, they would be used together. The end goal would be to intelligently design wind farms considering turbine design, layout, and control all working together.

Cf. the comment above. This result might very well be dictated by grid cost not being included.

This was responded to above in a comment from page 12.

**Page 20**

This is maybe a little contra-intuitive. Please comment on this finding.

The table is exactly the same data as Figures 12 and 14 presented differently. When the turbine is optimized in isolation, it is optimal to be large because it is always exposed to the high free stream wind speeds. In a farm, especially the small farm, the wind speeds that the turbine actually is exposed to are much lower from wake interference, so in these cases it is more optimal to be smaller. It happens that the baseline design is close to the optimal turbine for the closely spaced wind farm, which makes it better than the larger sequential optimization.

The paper includes a wind turbine cost model, which is crucial for this type of studies ... and thus a significant step forward compared to the work of e.g. Chen et al. However, regarding wind farm layout optimization another financial cost has a significant impact on the optimized layout. Please comment on how cable costs influence the conclusions of this paper. Is it e.g. possible that the large wind farm case would have been populated differently with turbines?

For a more complete picture, effects of individual wind turbine de-rating, with the purpose optimizing the wind farm production, must also be included. This approach offer potentially a multitude of 'different' turbines (in an operation context) in a wind farm ... without imposing additional (financial) costs.

A paragraph has been added to the conclusion recommending that future research includes cable costs, as they might affect the optimal turbine layout (especially in the big farms).

The second comment here, about active control, was responded to above in a comment from page 16.

**Page 21**

Please specify - what is exactly meant by 'implicit design variables' in this context?

This was clarified in the text a paragraph before equation 6.